**Brief Communication**

# SQANTI3: curation of long-read transcriptomes for accurate identification of known and novel isoforms

Francisco J. Pardo-Palacios [1,2,8], Angeles Arzalluz-Luque [1,2,8], Liudmyla Kondratova [3,4], Pedro Salguero[2], Jorge Mestre-Tomás [1], Rocío Amorín [4,5], Eva Estevan-Morió [1], Tianyuan Liu [1], Adalena Nanni[6], Lauren McIntyre[4,6], Elizabeth Tseng[7] & Ana Conesa [1] ✉

SQANTI3 is a tool designed for the quality control, curation and annotation of long-read transcript models obtained with third-generation sequencing technologies. Leveraging its annotation framework, SQANTI3 calculates quality descriptors of transcript models, junctions and transcript ends. With this information, potential artifacts can be identified and replaced with reliable sequences. Furthermore, the integrated functional annotation feature enables subsequent functional iso-transcriptomics analyses.

Long-read sequencing, driven by biotechnology companies such as Pacific Biosciences (PacBio) and Oxford Nanopore Technologies (ONT), was recognized as the method of the year 2022 by *Nature Methods*[1–3] for providing single-molecule reads spanning thousands of bases and advancing genomics and transcriptomics research. When applied to gene expression analysis, long-read RNA sequencing (lrRNA-seq) has the potential to capture full-length transcripts and elucidate isoform diversity in both normal and disease conditions[4–6]. Software tools have been developed for lrRNA-seq-based transcript identification[7–9], quantification[10,11], differential splicing analysis[6] and functional interpretation[12].

One of the most striking results in lrRNA-seq studies is the identification of thousands of novel transcripts, even in well-annotated genomes[5,10,13]. However, long-read technologies are error prone, and biases due to RNA degradation, library preparation issues and sequencing errors, as well as read mapping and transcript reconstruction inaccuracies, often lead to false transcript identification. Several studies have evaluated the accuracy of lrRNA-seq methods and algorithms[14–17]. These works have consistently highlighted significant disagreements between experimental and computational approaches at identifying transcripts from long-read data, especially for novel transcripts not present in the reference annotations. Disagreements involve the

annotation of splice junctions and the definition of transcription start sites (TSS) and transcription termination sites (TTS)[16], which are particularly difficult to discriminate from RNA degradation in the sequenced samples. Given the large number of novel isoforms reported by most lrRNA-seq studies, quality control and curation of the data are crucial steps in long read-based transcriptome definition.

We hereby present SQANTI3, a tool for the evaluation of long-read transcript models used as an evaluation engine in the LRGASP (Long-read RNA-seq Genome Annotation Assessment Project)[16]. SQANTI3 builds on SQANTI[18], a widely used tool for quality control of lrRNA-seq data (a comparison of SQANTI and SQANTI3 functionality is given in Supplementary Note 1).

The SQANTI3 workflow consists of three modules (Fig. 1a). First, quality control (QC) classifies long-read transcript models according to SQANTI3 structural categories, which consist of the SQANTI splice junction-based transcript classes: full-splice-match (FSM); incomplete-splice-match (ISM); novel-in-catalog (NIC); novel-not-in-catalog (NNC); antisense; fusion; genic genomic; and intergenic (Fig. 1b); and novel subcategories based on TSS and TTS annotations (Fig. 1c).

Of these, 'reference match' is defined as an FSM transcript in which both the 3′ and 5′ ends are within 50 bp of the reference transcript's

[1]Institute for Integrative Systems Biology, Spanish National Research Council, Paterna, Valencia, Spain. [2]Department of Applied Statistics and Operational Research, and Quality, Universitat Politècnica de València, Valencia, Valencia, Spain. [3]Horticultural Sciences Department, University of Florida, Gainesville, FL, USA. [4]Genetics Institute, University of Florida, Gainesville, FL, USA. [5]Department of Microbiology and Cell Science, University of Florida, Gainesville, FL, USA. [6]Department of Molecular Genetics and Microbiology, University of Florida, Gainesville, FL, USA. [7]Pacific Biosciences, Menlo Park, CA, USA. [8]These authors contributed equally: Francisco J. Pardo-Palacios, Angeles Arzalluz-Luque. ✉e-mail: ana.conesa@csic.es

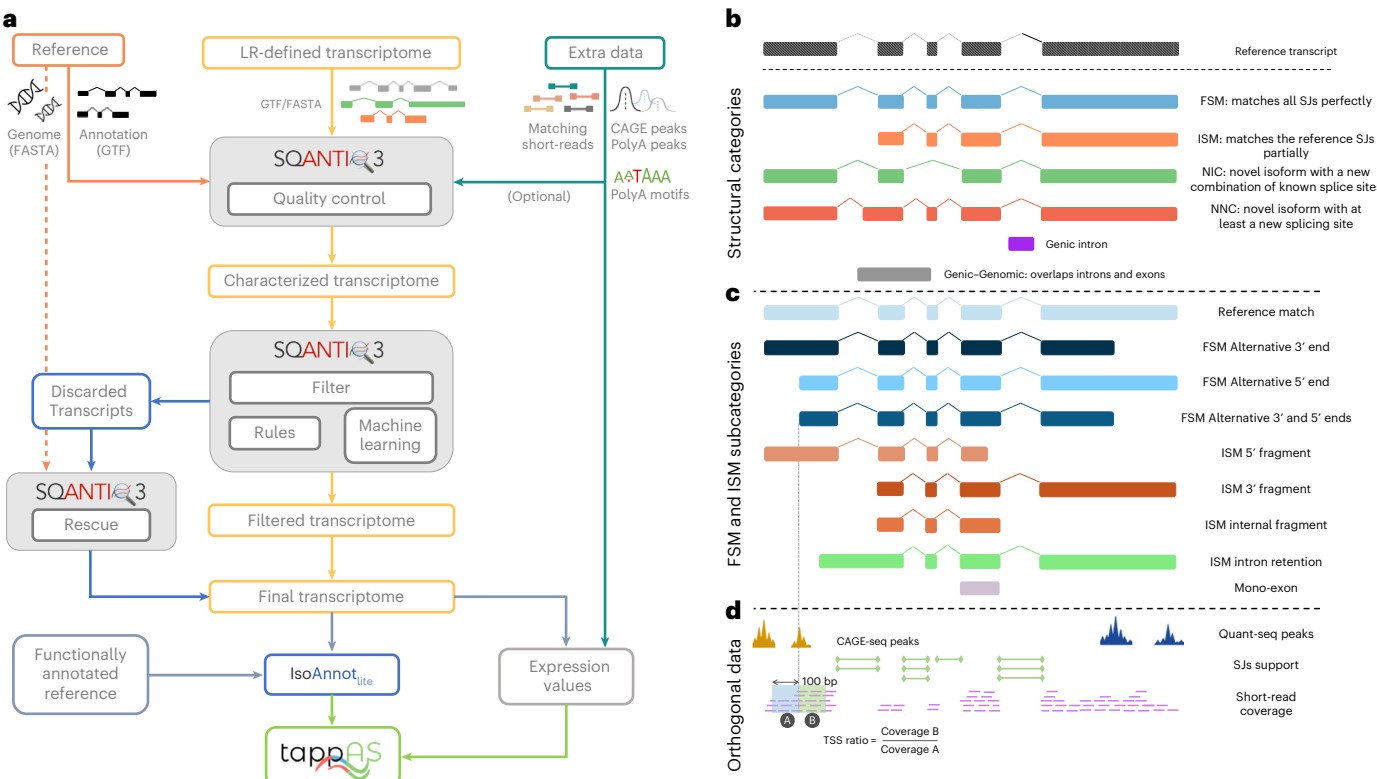

**Fig. 1 | Overview of SQANTI3. a**, SQANTI3 workflow. **b**, Main SQANTI structural categories for transcript models of known genes. **c**, SQANTI3 subcategories for FSM and ISM transcripts. **d**, Orthogonal data features processed by SQANTI3 QC. LR, long read; SJ, splice junction.

TTS and TSS, respectively, and can be considered as perfect hits when comparing lrRNA-seq transcript models to the reference. Larger variations at each or both ends are captured by the alternative 3′ and 5′ subcategories. Similarly, SQANTI3 divides the ISM class based on whether missing splice junctions are located at the 5′ (3′ fragment), 3′ (5′ fragment), or both ends (internal fragment). If splice junctions are lost due to intron retention, these transcripts are labeled as Intron Retention. In total, SQANTI3 defines 22 structural categories and subcategories (complete definitions are given in the 'Code availability' section). The SQANTI3 QC module additionally computes up to 48 transcript-level and 18 junction-level quality features. These QC indicators describe, among other characteristics, the presence of noncanonical splice junctions, the existence of intrapriming, and potential reverse transcriptase switching events. Moreover, SQANTI3 QC can assess the reliability of TSS and TTS annotations by processing complementary data such as CAGE (capped analysis of gene expression)[19], Quant-seq[20] or other genomic region data (Fig. 1d) and calculating the distance and overlap between transcript ends and these regions. The software also integrates the processing of short-read data to support splice junction annotations and introduces the TSS ratio metric, that is, the ratio of short-read coverage downstream to that upstream of the TSS. 5′ end-degraded transcripts are expected to display uniform coverage on both sides of the TSS (that is, TSS ratio ≈ 1), while a true TSS is expected to have significantly lower upstream coverage, resulting in a TSS ratio > 1. SQANTI3 QC main output files are the classification table, which includes all quality features for each annotated transcript model, and the QC report, which provides multiple summary statistics and diagnostic plots that help identify biases and false positives.

The second module in the SQANTI3 workflow is artifact filtering based on QC descriptors. Two modes are available. The machine learning mode builds a random forest artifact classifier using QC features as predictive variables and a set of true- and false-positive transcripts for model training. Given that the definition of a training set might not

always be feasible, SQANTI3 alternatively offers a rules mode, in which (sub)category-level exclusion criteria are manually defined by the user. After filtering, however, known genes may be completely removed when all of their transcripts are classified as artifacts. To mitigate the risk of excluding transcripts and genes that show evidence of expression, SQANTI3 includes a third rescue module in which artifacts are assigned to the most suitable reference or long-read transcript model by applying a two-step process that recovers reference transcripts for discarded FSM transcripts, and orthogonal data-supported alternatives for ISM, NIC and NNC (Extended Data Fig. 1). Additionally, SQANTI3 enables functional annotation through the IsoAnnotLite tool, which maps pre-annotated functional domains, motifs and sites onto the long-read transcript models using a genomic coordinate-matching strategy (Methods). The result is a sample-specific transcriptome in which isoform expression changes can be interrogated for their potential functional implications. An example of complete functional iso-transcriptomics analysis enabled by SQANTI3 is given in Supplementary Note 2.

To demonstrate SQANTI3, we retrieved PacBio complementary DNA data from the human WTC11 cell line from LRGASP[16] for which Illumina, CAGE-seq and Quant-seq data were also available. Transcript models were generated from long reads using IsoSeq3 (Methods). In total, 228,379 transcript models were constructed, with 209,220 (91.6%) belonging to 17,467 known genes (Fig. 2a). Nearly one-third of all transcripts were classified as ISM (67,804; 29.69%), while 37% of them were categorized as novel isoforms of annotated genes (48,878 NIC and 35,743 NNC). Only 56,795 transcripts (24.87%) were annotated as FSM, a relatively small fraction considering that WTC11 is a well-studied human cell line. This result may reflect reference catalog incompleteness, or inaccuracies in the long-read technology or the transcript reconstruction algorithm. These biases can be further investigated using SQANTI3.

Given that splice junction novelty was previously discussed and evaluated using NIC and NNC SQANTI classes[18], in this study we show

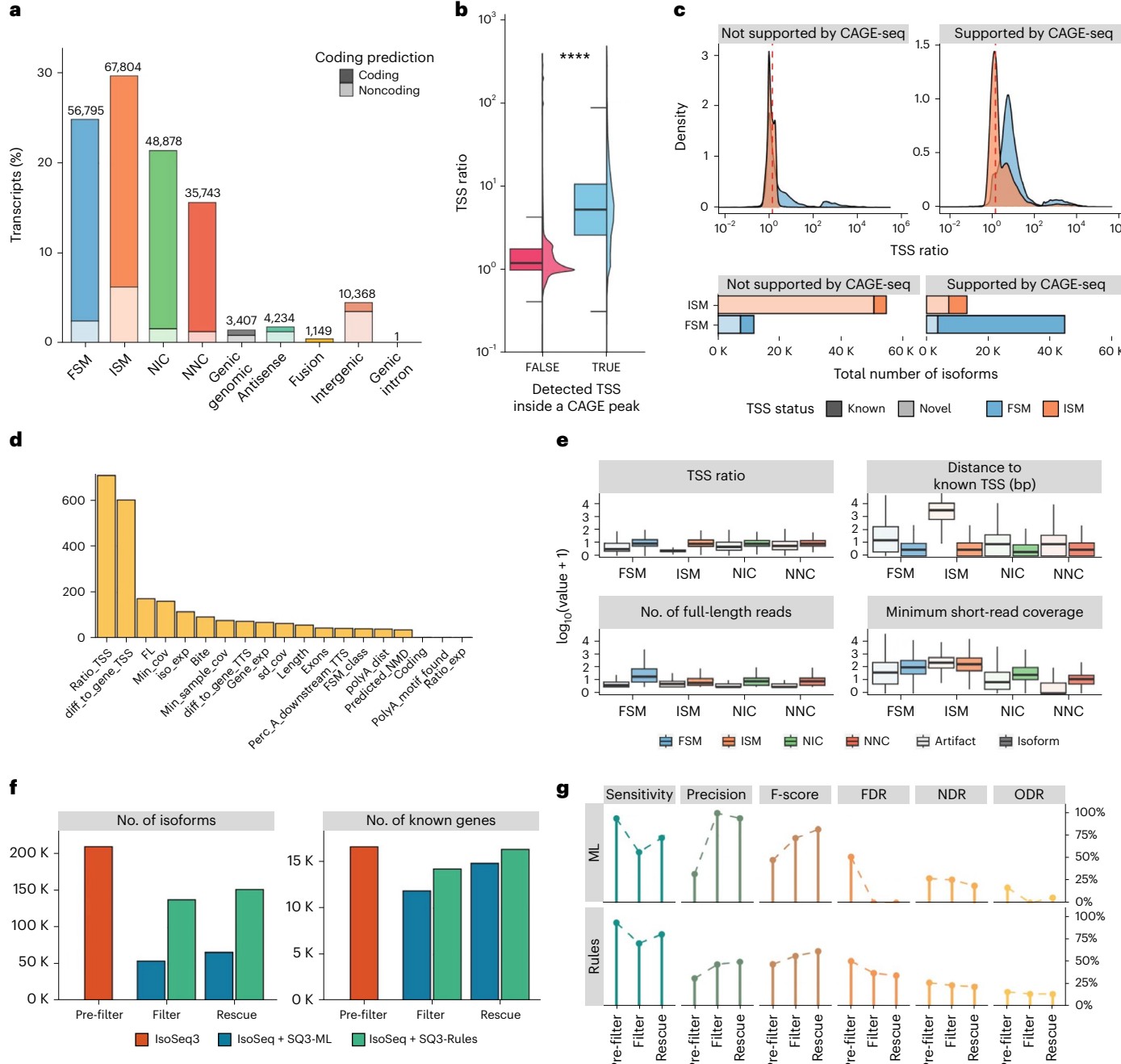

**Fig. 2 | Validation of SQANTI3 features. a**, Distribution of transcript models by SQANTI3 structural categories in the WTC11 IsoSeq3-defined transcriptome. **b**, Differences in TSS ratio between TSS supported and not supported by CAGE-seq data (****$P = 2 \times 10^{-16}$, two-sided Wilcoxon test). **c**, Uneven support of CAGE-seq data for FSM and ISM transcript models, with supported TSS usually having a TSS ratio > 1.5 (red dashed line), particularly if they are also known TSS. **d**, Variable importance of SQANTI3 descriptors in the machine learning (ML) filter for different input scenarios, obtained after training the random forest model on different true-positive (TP) sets. **e**, Distribution of values for the top machine learning filter variables (ranked by random forest classifier importance) for isoforms and artifacts across the FSM (35,446 isoforms, 21,079 artifacts), ISM

(2,573 isoforms, 65,114 artifacts), NIC (15,504 isoforms, 33,367 artifacts) and NNC (1,556 isoforms, 34,182 artifacts) structural categories. **f**, Variation in the number of genes and transcripts after filter and rescue using sample-specific orthogonal data with rules and machine learning approaches. **g**, Performance metrics according to SIRV detection at each step of the SQANTI3 pipeline using rules and machine learning approaches. FDR, false discovery rate; NDR, novel detection rate; ODR, over-annotation detection rate For all boxplots in this figure, the middle line represents the median, the ends of the box represent the 25th (quartile 1) and 75th (quartile 3) percentiles, and the whiskers represent the minimum (quartile 1 minus 1.5-fold the interquartile range (IQR)) and the maximum (quartile 3 plus 1.5-fold the IQR). The half-violin plots show the density distribution of values.

how SQANTI3 improves the description of 3′ and 5′ end variability of lrRNA-seq transcript models. First, the reliability of the TSS ratio metric for accurate TSS identification was evaluated by comparing SQANTI3 ratios with empirical evidence from CAGE-seq data. Among all identified TSS, those with additional support from sample-specific

CAGE-seq had significantly higher TSS ratios (Fig. 2b, $P = 2 \times 10^{-16}$, Wilcoxon test). Specifically, 88.2% of transcripts supported by CAGE-seq had a TSS ratio greater than 1.5, which was used as the cut-off value for short read-based TSS support in downstream analyses (Methods). Overall, we observed substantial agreement between our

three 5′ end metrics, namely, same-gene TSS annotation, CAGE-peak, and TSS ratio, in terms of TSS support (Extended Data Fig. 2).

Next, we evaluated the consistency of SQANTI3 5′ end descriptors across SQANTI3 structural categories. As expected, the vast majority of TSS in FSM transcripts had both CAGE-seq support and TSS ratio > 1.5, whereas this was not the case for ISM transcripts, confirming that these metrics recapitulate the reference annotation (Fig. 2c). Interestingly, 11.2% (7,599) of ISM transcripts had TSS with both CAGE peak overlap and a TSS ratio above 1.5, with two-thirds of them also matching a same-gene annotated TSS. Furthermore, 4,591 (8%) FSM transcripts had a low TSS ratio and lacked support from the reference annotation and CAGE-seq peaks (Fig. 2c). These findings suggest that SQANTI3 5′ end features constitute valuable tools for verifying novel alternative TSS both in the ISM and FSM categories.

Regarding 3′ end diversity, SQANTI3 QC was used to determine the support for detected TTS through three distinct types of evidence: Quant-seq data, the PolyASite database annotation, and the presence of a polyadenylation (polyA) motif in the final 50 bp of the transcript sequence. When examining the correlation between them, the majority of transcripts (165,612) had a TTS supported by all three sources of evidence. This was followed by instances in which a polyA motif was identified near an annotated polyA site (25,696). In only 1% of cases (2,269), the Quant-seq peak overlapped with an unannotated polyA site (Extended Data Fig. 3). Furthermore, the identified motifs were concentrated at a distance of 16–18 bp from the end of the transcript, aligning with available experimental evidence[21]. This pattern held even for transcripts that were not validated with Quant-seq (Extended Data Fig. 4). These results suggest that long-read sequencing methods have higher sensitivity in detecting alternative 3′ ends compared with Quant-seq, possibly because Quant-seq necessitates high expression to call a polyA site. Transcripts that were likely to be a product of intrapriming (defined as 60% of As downstream of the TTS[18]), rarely contained a polyA motif or the polyA was located closer to the 3′ end than expected (Extended Data Fig. 5). Overall, these findings further validate the detection of polyA motifs as a reliable indicator of a bona fide TTS.

Furthermore, an in-depth analysis of transcript end variability using SQANTI3 subcategories suggests that unsupported ISM transcripts are more likely to be 5′ degradation products (Supplementary Note 3). Interestingly, subcategory analysis also showed that an important fraction of FSM transcripts have notable differences at 3′ and 5′ ends with respect to the reference annotation, with orthogonal data supporting many of the alternative TTS and TSS, indicating that although FSM transcripts are usually regarded as 'known transcripts', alternative transcripts are also present in this transcript category (Supplementary Note 3).

SQANTI3 QC analyses suggest that a combination of artifacts and true novel transcripts populate the IsoSeq3 transcriptome. The SQANTI3 filter module identifies and removes potential artifacts using either the machine learning or the rules mode. In the machine learning filter, the trained classifier is used to obtain transcript-level artifact probabilities, reporting the QC features that have contributed the most to the discrimination of artifacts and transcripts during model training and their values for true or artifact transcripts in each SQANTI3 category (Fig. 2d,e and Supplementary Note 4). Previous work demonstrated the efficiency of the machine learning filter in removing transcript models with erroneous junction definitions (that is, NIC and NNC) and other low-abundance transcript classes[18]. Here, we show that the improved SQANTI3 machine learning filter, which incorporates 3′ and 5′ end descriptors, can also curate FSM and ISM transcript models (Fig. 2e and Extended Data Fig. 6). Alternatively, for users requiring precise control over data filtering, the rules mode enables ad hoc definition of transcript inclusion criteria. Importantly, both modes in the filter module enable strategic adaptation to the type and amount of available supporting data. Supplementary Note 4 describes the performance of the SQANTI3 filter module in three scenarios with varying data availability, that is, a low-input setting (equivalent to the previous SQANTI machine learning filter), and two high-input settings, using public database information or orthogonal same-sample data (Methods). We observed that the high-input setting permitted more thorough control of transcript end quality.

Filtering resulted in the removal of a large number of transcript models for many data scenarios (Fig. 2f). As a consequence, a substantial number of genes were completely excluded because none of the associated transcripts passed the filtering criteria, despite evidence of expression provided by long reads mapping to these loci. SQANTI3 rescue was designed to mitigate these losses by identifying the most likely transcript model for each discarded artifact. During the rescue step, discarded ISM, NIC and NNC transcripts are first mapped to the reference transcriptome, which results in the identification of multiple potential replacement transcripts per artifact. When applying this strategy to machine learning high-input filtered data, most of these rescue candidates belonged to the reference transcriptome (Extended Data Fig. 6). After successive steps to validate the candidates and reduce redundancy (Methods), 11,599 artifacts (89%) were re-incorporated into the transcriptome, 94.1% of which were assigned to a reference transcript and 5.9% were assigned to a novel long read-defined transcript model, with a total of 2,884 new genes being added. Notably, only 11% of artifacts remained unassigned to a suitable replacement transcript after the rescue process (Fig. 2f). Hence, a large number of artifacts could be matched to already-identified transcript models, suggesting that an important fraction of detected artifacts may be the consequence of errors that affect correct transcript detection. Moreover, rescued genes and transcripts had consistently higher expression and functional scores (TRIFID scores) in the APPRIS database[22] than those that remained excluded (Extended Data Fig. 7). In summary, these results demonstrate that the SQANTI3 rescue strategy can recover functionally relevant known genes and transcripts that would otherwise be discarded due to lrRNA-seq limitations. The SQANTI3 QC, filter and rescue modules are also effective in characterizing and curating transcriptomes generated using other types of lrRNA-seq data (for example, direct RNA sequencing[23]) and analysis tools (for example, TALON[7]; Supplementary Note 5).

To validate the ability of SQANTI3 to obtain a correctly curated transcriptome, we benefited from the usage of spike-in RNA variants (SIRVs) in the LRGASP WTC11 PacBio dataset. We modified the SIRV reference annotation to simulate two real-life scenarios leading to transcript artifacts: an over-annotated false-positive scenario, in which 39 falsely annotated SIRVs-like models were included in the transcriptome; and a novel true-positive scenario, in which 26 true transcripts were removed from the reference, even though they were still present in the control RNA mix (Methods). Although the initial IsoSeq3 reconstruction yielded almost perfect sensitivity (94%), precision was low (32%) (Fig. 2g). Both machine learning and rules filtering improved precision and lowered sensitivity, while performing the rescue step restored sensitivity values without significantly affecting precision (Fig. 2g). F-scores showed that overall performance steadily improved after every step of the SQANTI3 curation pipeline, with the highest F-score being observed when using the machine learning filter. In agreement, the false discovery rate consistently decreased after filtering and rescue, especially when the machine learning filter was applied (Fig. 2g). Rules filter-based strategies were less effective at removing false positives, which resulted in an almost constant over-annotation detection rate, whereas machine learning-based approaches decreased the over-annotation detection rate effectively. Finally, both filters yielded a constant novel detection rate (Fig. 2g). These results show the power of SQANTI3-based curation to validate known and novel transcripts and effectively leverage the potential of long-read methods to discover new transcripts.

In conclusion, this work shows that SQANTI3 is a comprehensive and flexible tool for the structural characterization and quality control of lrRNA-seq-derived transcriptomes. It can integrate orthogonal

data to improve the accuracy of transcript models and provides a range of filtering options to accommodate different research goals. For genome annotation, we recommend using extensive orthogonal data and applying machine learning-based filtering to obtain a set of high-confidence transcript models. In other applications that seek to detect rare novel transcripts, more lenient filtering may be applied to allow for discovery, especially when follow-up validations are planned. Finally, for isoform-resolved differential expression studies, filtering based on consistent detection across samples is advisable. With the ability to curate novel transcripts and provide functional annotation, SQANTI3 is an essential tool in the long-read transcriptomics field.

## Online content

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

## Methods

### Novel features in SQANTI3

**SQANTI3 quality control module.** The quality control module is the cornerstone of the SQANTI3 pipeline. It is designed to characterize transcriptomes built using lrRNA-seq data and make QC decisions according to the purpose of the study. Combining Python and R scripts, SQANTI3 QC compares the de novo transcriptome against a reference annotation and outputs an easy-to-explore classification table file. This file feeds the graphical output, is used as input for the filter module, and is generated for the reference annotation when running SQANTI3 rescue.

*New subcategories for full splice and incomplete splice matches.* SQANTI3 QC expands the long-read transcript model classification scheme defined by SQANTI[18] by characterizing the variability at the ends of the transcript models to create subcategories for reference-associated transcript categories, that is, FSM and ISM.

FSM can be directly associated with known transcript models based on their splice junction, but differences at their ends are not negligible. When differences are small, meaning that the FSM transcript ends match the reference with a difference of ≤50 bp upstream or downstream from the annotated TSS and TTS, the transcript models are assigned to the reference match subcategory. Conversely, transcripts with a ≥50 bp distance to the annotated TSS or TTS are considered 'alternative', and annotated to one of the three additional subcategories: alternative 5′end; alternative 3′end; or alternative 5′or 3′ends. For ISM, which are fragments of annotated splice junction combinations, subcategories are defined based on the missing fraction of the known transcript. ISM can therefore be classified as 3′fragments, 5′fragments, or internal fragments, depending on whether they lost exons at in their 5′, 3′, or both ends. An additional subcategory, ISM with intron retention, was defined to classify transcripts for which the loss of a splice junction in the reference is due to an intron retention event. Finally, the mono-exon subcategory was included for both structural categories given that, despite their lack of splice junctions, they can be associated with known transcripts by overlap. Specifically, if a mono-exon transcript overlaps a mono-exonic reference, it will be classified as FSM, whereas if the reference is multi-exonic and the query sequence lies within the boundaries of an annotated exon, it will be considered an ISM. NIC and NNC categories remain the same as they were defined in SQANTI[18].

*Integration of evidence around 5′ and 3′ ends.* In addition to short- and long-read data for coverage- and expression-based validation, SQANTI3 QC can make use of additional data to generate metrics related to TSS and TTS support, namely CAGE, Quant-seq data or other region-based sources of information providing TSS and TTS evidence. Specifically, the QC module accepts BED (browser extensible data) files containing the genomic coordinates of specific regions called (for example, CAGE peaks) as input. The overlap between each TSS and TTS reported in the lrRNA-seq transcriptome and the supplied regions are verified, flagging cases where transcript ends fall inside a peak. Additionally, the distance between the TSS or TTS and the middle point of the closest peak is computed. Only peaks upstream of the TSS or downstream of the TTS are interrogated for this purpose.

*Processing of short-read data.* To facilitate the integration of matching short-read data, SQANTI3 has been upgraded to accept FASTQ Illumina data and run STAR (spliced transcripts alignment to a reference)[24] and Kallisto[25] internally for mapping and quantification purposes (Supplementary Methods).

First, a genome index is created and short reads are mapped to identify individual splice junctions using STAR. The reference annotation is not used in this process, to make splice junction identification completely independent from prior annotations. Mapping parameters used in STAR are adapted from the ENCODE-DCC protocol for RNA-seq (https://github.com/ENCODE-DCC/rna-seq-pipeline/), however, to improve the detection and quantification of novel splice junctions, the `-twopassMode` option is activated[26]. After running STAR, an SJ.out.tab and a BAM (binary alignment map) file are generated for each replicate. The SJ.out.tab file contains the short-read quantification of splice junctions defined by long reads, whereas the BAM file is used to calculate the novel TSS ratio metric. Taking the genomic position of all of the long read-defined TSS in the query annotation, two BED files containing the 100 bp regions downstream (inside the first exon) and upstream (outside the first exon) of the TSS are created. Using BEDTools[27], short-read coverage across both genomic segments is measured and the TSS ratio is calculated using the following formula:

$$\text{TSS ratio} = \frac{\text{coverage inside} + 0.01}{\text{coverage outside} + 0.01}$$

For each TSS, SQANTI3 QC computes as many ratio values as short-read sequencing replicates are provided, although only the highest ratio is retained by default. Alternatively, users can select other metrics for TSS ratio aggregation across samples, namely the mean, median and third quartile. Details are available in the SQANTI3 software documentation (see 'Code availability').

For transcript model quantification, SQANTI3 runs Kallisto internally when short-read paired-end data are provided. First, an index is built using the transcript sequences extracted from the query annotation and the reference genome. Then, short reads are pseudoaligned to these genome-corrected sequences to quantify them using `-bootstrap-samples 100` as the only non-default parameter.

**SQANTI3 filter module.** *Machine learning-based filter.* SQANTI3 substantially improves the SQANTI machine learning-based filter (machine learning filter)[18]. In brief, the filter is based on the training of a random forest classifier to calculate the probability that a given transcript model is an isoform or an artifact. The training process is based on SQANTI3 QC attributes and on the selection of true-positive and true-negative transcript sets that will show differences in these attributes.

In SQANTI3, the definition of true-positive and true-negative sets has been modified to enable either user-defined transcript lists for model training or an automated selection of true-positive and true-negative transcripts based on SQANTI3 categories and subcategories when no true-positive and true-negative lists are supplied. True-positive and true-negative lists should each contain at least 250 transcripts for the filter to run, and lower sizes have been banned to prevent unstable behavior in model training. By default, NNC noncanonical isoforms are taken as true negative, whereas the Reference Match subcategory is used as the true positive, because these transcript models have both reference-supported junctions and ends. In these cases, attributes related to junction type (canonical or noncanonical) and to the distance to the reference TSS or TTS are automatically excluded to prevent biases and overfitting. In addition, the size of these sets is now automatically balanced, downsizing the largest to match the number of transcripts in the smallest training list. The maximum training set size can be adjusted by the user to alleviate the computational burden of model training, even in cases in which true positive and true negative are automatically defined. Relative to filtering criteria, the SQANTI3 implementation of the machine learning filter has been designed to be more stringent on the isoform than on the artifact condition, requiring that the probability that the transcript is a true isoform be ≥0.7 by default. This threshold, nevertheless, can be modified by the user, and the probability distribution generated by the classifier is included as part of the machine learning filter report to assist this decision. Finally, the machine learning filter now allows users to force the inclusion or exclusion of some specific isoform groups. For FSM transcripts, users may indicate that all of them

should be included as true isoforms in the filtered transcriptome. In addition, given that mono-exonic transcripts are not evaluated during machine learning filtering due to the lack of junction-related attributes, these can be automatically removed from the transcriptome if desired; otherwise, they will not be subject to filtering.

A key improvement to the machine learning filter in SQANTI3 is the use of the novel TSS and TTS validation metrics and additional data incorporated during QC for classifier training, namely CAGE-seq peaks, the short read-based TSS ratio metric, polyA motif information, and Quant-seq peaks. As a result, the filter can now detect artifacts belonging to the FSM and ISM categories, which were automatically flagged as true isoforms in the original release. By default, all available variables in the SQANTI3 classification file are used for model training, except those related to genome structure (for example, chromosome or strand), to the associated reference transcript or gene, and the SQANTI categories and subcategories. Moreover, the SQANTI3 machine learning filter includes the possibility to exclude variables, to prevent overfitting in cases in which one or more of these variables have served as criteria for true-positive and true-negative set definitions. Additional details, including how to run the machine learning filter, are available at the SQANTI3 wiki (see 'Code availability').

*Rules filter.* To make the filter module more flexible and accommodate cases in which the definition of true-positive and true-negative sets is not possible, a rules-based strategy for artifact removal has also been included in SQANTI3. A JSON file[28] is used to specify the characteristics that make an isoform reliable.

The JSON file is structured in two different levels of hierarchy: rules and requisites. A rule is made of one or more requisites, all of which must be fulfilled for an entry to be considered a true transcript. This means that requisites will be evaluated as AND in terms of logical operators. If different rules (that is, sets of requisites) are defined for the same structural category, they will be treated independently from one another. In that case, to pass the filter, transcripts will need to pass at least one of these independent rules, meaning that rules will be evaluated as OR in terms of logical operators. Rules can be set for any numeric or categorical column in the classification file. Numeric value thresholds can be defined as a (closed) interval of accepted values or by setting only the lower limit. For categorical QC attributes, users may define one or more acceptable levels. More details and examples of how to correctly define the rules are available at the SQANTI3 wiki (see 'Code availability').

The default filter includes two sets of rules. FSM transcripts are removed solely based on intrapriming, that is, if they have ≥60% of As in the sequence immediately downstream of the TTS of the transcripts. Transcripts from other categories are required to be negative for intrapriming and reverse transcriptase switching, as well as to have all of their junctions supported by at least three short reads or to have only canonical junctions. However, the JSON-based rule definition enables users to apply ad hoc filtering criteria to each SQANTI3 category, which may involve thresholds for multiple QC attributes and any type of available orthogonal data.

**SQANTI3 rescue module.** The SQANTI3 rescue algorithm was conceived to be run after removing potential artifacts from the transcriptome using the SQANTI3 filter module with the goal of avoiding the loss of transcripts and genes that constitute part of the transcriptional signal, but for which a correct transcript model could not be generated when processing the long-read data. In practice, this is done by selecting a replacement transcript from the reference that is ultimately added to the set of long read-defined, filter-passing transcript models to generate an expanded, final version of the transcriptome. This strategy is based on two principles: consistent quality, meaning that rescued transcript models should meet the QC criteria of the filtering settings used to call isoforms, and non-redundancy, meaning that when the

identified replacement transcript for a given artifact is already part of the filtered transcriptome, no transcript model is added. The SQANTI3 rescue algorithm operates in four steps.

The first step in this module, hereby referred to as automatic rescue, applies to FSM artifacts, which may originate when TSS or TTS validation fails. In this case, reference transcripts associated with at least one removed FSM transcript are automatically added to the transcriptome. When multiple removed FSM transcripts have the same associated reference transcript but different TSS, the reference sequence is added only once. Artifact transcripts from the ISM, NIC and NNC categories are considered as rescue candidates and will continue to be analyzed by the rescue pipeline. ISM artifacts are considered only if they do not have an FSM counterpart associated with the same reference transcript.

Second, a group of rescue targets, that is, potential replacement transcripts for the rescue candidates, is defined. This includes all long read-defined and reference transcript models for which at least one same-gene rescue target was found. Matches between each rescue target and its same-gene candidates are next found by mapping candidate sequences to targets, a process known as rescue by mapping. To achieve this, minimap2 (ref. 29) is run in the long-read alignment mode using the `-a` parameter, combined with the `-x map-hifi` (that is, high-fidelity read alignment) preset option to reflect the accuracy of the processed transcript sequences. Secondary alignments are allowed and set to the default number of 6 to enable multiple mapping hits to be reported per candidate. This process yields a series of alignments that pair each rescue candidate to multiple possible targets, pairs that are hereby referred to as mapping hits.

Third, to ensure that reference transcriptome targets included in the rescue process comply with the same quality requirements as long-read targets and to minimize the risk of retrieving nonsample-specific transcripts, SQANTI3 QC and filter modules are run on the reference transcriptome, supplying the same additional data and quality criteria as used for the long-reads transcript models. Finally, the rescue-by-mapping process is completed by applying a series of criteria to select suitable targets for inclusion in the final transcriptome. First, mapping hits are disregarded if the rescue target did not pass the filter, be it machine learning or rules. When multiple mapping hits per rescue candidate exist that pass the filter in the machine learning strategy, the target transcript with the highest machine learning filter probability (long-read or reference) will be selected for rescue. If multiple hits pass the rules filter, all targets will be considered, given that there are no further refinement criteria available in this case. If the best match target is a long read-defined transcript model that is already included in the transcriptome, no further action is performed because the artifact is already represented in the dataset. The remaining reference targets are then evaluated for potential redundancy against the curated transcriptome, and added only if not already present.

By default, the rescue module runs only the automatic rescue step, enabling the recovery of high-confidence reference transcripts using FSM evidence only. However, users can modify this behavior if they wish to perform a more comprehensive rescue using ISM, NIC and NNC artifacts to find matching replacement transcripts. More information is available at the SQANTI3 wiki (see 'Code availability').

**IsoAnnotLite and tappAS integration.** IsoAnnotLite is a Python script for the transfer of isoform-level functional feature annotations from an existing tappAS annotation GFF3 (general feature format version 3) file to long read-defined transcripts processed using SQANTI3 QC. During this process, long read-defined transcripts, that is, feature acceptors, receive annotations from transcripts that are already annotated with functional information and therefore act as feature donors. The resulting GFF3 file is compatible with the tappAS software for isoform-level functional analysis. In the case that a reference GFF3 annotation file is not provided, the output transcriptome will be formatted into a

tappAS-compatible GFF3 file including only structural information, enabling quantitative but not functional analysis within tappAS.

The script uses the classification, junction and GTF file generated by SQANTI3 QC, and can be run simultaneously with the QC script by supplying the `-isoAnnotLite` flag and providing a tappAS GFF3 functional annotation file via the `-gff3` argument. IsoAnnotLite is executed with `-novel` as the only non-default parameter to force all transcripts to be treated as novel transcripts, meaning that each long-read transcript or feature acceptor is annotated using functional information from all feature donors belonging to the same gene in the reference GFF3. When the `-novel` argument is not supplied, the non-novel acceptor transcript will receive only annotations from donor transcripts with matching identification numbers. More information on IsoAnnotLite parameters and their effect on the annotation process is available in the SQANTI3 documentation (see 'Code availability').

The IsoAnnotLite algorithm includes the following steps. First, positional information for feature acceptors, that is, transcripts in the lrRNA-seq transcriptome, is converted to genome positions using the information in the SQANTI3 QC output. Similarly, transcript-level functional feature positions from the donor transcripts are transformed into genomic coordinates using the information in the reference GFF3 file. Next, functional features are transferred across transcript models by matching genomic positions, that is, features from donor transcripts for which the genomic positions span a feature acceptor will be annotated as belonging to that transcript. It should be noted that different transfer rules have been implemented depending on the type of feature that is being handled. To transfer features of the untranslated region, genomic feature positions must be inside the transcript's exons and outside its coding sequence (CDS) region. For CDS transcript features (namely transcript-level features situated in the coding region), the feature must be contained in the acceptor transcript's exons as well as inside the CDS region; and if a feature has start and end positions situated in different exons, the end and the start of the exons for the donor and acceptor transcripts must be the same for IsoAnnotLite to transfer the feature. For protein features, the donor and acceptor transcripts are first verified to be coding and have the same CDS. If all CDS exons are the same for both transcripts, all protein features are automatically transferred. If not, IsoAnnotLite requires the genomic positions of at least one CDS exon to be a partial match, that is, for the feature donor and acceptor to share part of one exon in the transcript's CDS. If at least one CDS genomic region overlaps between both transcripts, IsoAnnotLite checks for protein features that fall inside that region and can therefore be transferred. For gene-level characteristics (for example, Gene Ontology terms), information is always transferred across matching gene identification numbers. Finally, IsoAnnotLite verifies whether the same feature has been transferred from multiple donor transcripts to the same acceptor, and performs the removal of duplicated annotations.

### Data

The WTC11 cell line is an induced pluripotent stem cell line derived from human fibroblasts, often used as a model for cell differentiation[30]. The data used in this paper were generated as part of LRGASP[16], in which this cell line was deeply sequenced using different technologies. We used only the cDNA PacBio data for reconstructing transcript models. Raw data used in this study (subreads) are accessible through the ENCODE database, under experiment accession ENCSR507JOF. Additionally, raw short-read data from the same samples were retrieved from ENCODE experiment accession ENCSR673UKZ. In both cases, sequenced RNA samples included Lexogen's Spike-In RNA Variants SIRV-Set 4 (cat. no. 141). This included 69 short SIRVs, 15 long SIRVs and 92 mono-exonic ERCC (External RNA Controls Consortium) transcripts. Short SIRVs (191 bp–2.5 kb) are designed to reproduce different splicing patterns with respect to their reference gene, creating a multi-isoform scenario. In contrast, long SIRVs (4–12 kb) do not contain splicing. The 69 short SIRVs resemble multi-exonic genes with alternatively spliced isoforms,

and were used as ground truth to evaluate performance after each step in the SQANTI3 pipeline.

CAGE-seq peak data for the WTC11 cell line were obtained from LRGASP (GEO accession GSE185917), while Quant-seq data were downloaded from ENCODE experiment ENCSR322MWL. Reads had been processed to obtain a collection of sample-specific peaks, and these were filtered to include only peaks found in at least two replicates, resulting in 46,722 CAGE-seq and 45,813 Quant-seq peaks[16]. Reference annotations of TSS and TTS were obtained from the Reference Transcription Starting Sites[31] (refTSS, v3.1) database and from the PolyASite database[32] (v2.0), respectively. A list of common human polyA motifs was obtained from ref. 21.

### Data processing

**Transcriptome reconstruction with IsoSeq3.** The PacBio cDNA lrRNA-Seq datasets used in the present study, that is, WTC11 and H1-DE (Supplementary Note 2), were processed using the IsoSeq3 software (v3.4.0) for de novo long-read transcriptome reconstruction, provided by PacBio (https://isoseq.how/). We followed the recommended pipeline by PacBio to build a transcriptome starting from subreads:

1. For each replicate in any given dataset, the `ccs` function was run using default parameters and setting `-min-rq 0.9` to define the minimum predicted accuracy.
2. The `lima` function was next run using the `-peak-guess` and `-isoseq` arguments and default parameters. This enables the identification of primers and the removal of chimeric reads.
3. Primer sequences were then trimmed using IsoSeq3 `refine`, with the `-require-polya` argument to keep only reads in which both primers and a polyA tail were identified, that is, full-length non-chimeric (FLNC) reads.
4. Replicates (and samples) were pooled together for the IsoSeq3 `cluster` step.
5. Transcript collapse was then performed to minimize transcript model redundancy using cDNA Cupcake (https://github.com/Magdoll/cDNA_Cupcake/). After mapping transcripts to the genome with minimap2 (ref. 29), the `collapse_isoforms_by_sam.py` script was run using the `-dun-merge-5-shorter` flag to prevent the removal of alternative TSS.

For further details on the exact code used to run IsoSeq3, see Supplementary Methods.

**Running SQANTI3 QC.** SQANTI3 QC was run using the human GENCODE annotation (v39) and default parameters for all three datasets. The different types and sources of orthogonal data used are reported in Supplementary Note 4.

Fisher's exact test was used to assess the capability of the TSS ratio and polyA motifs to recapitulate the information provided by experimental data. This was done by building contingency tables in which transcripts constituted the counted events, the columns represented whether or not transcripts had support by CAGE-seq or Quant-seq peaks, and the rows represented whether transcripts were validated by a TSS ratio > 1.5 or by the presence of a polyA motif, respectively.

**Running SQANTI3 filter.** *Rules filter.* For the WTC11 dataset, the rules filter was defined as follows. The 5′ ends were considered valid if they overlapped a CAGE-Seq peak OR an annotated refTSS site; the distance to any other annotated TSS in the same gene was less than 50 bp; or they had a TSS ratio > 1.5. Similarly, 3′ ends were accepted if they were supported by Quant-seq data OR by polyA site annotation; the distance to any other annotated TTS was less than 50 bp; or there was a canonical polyA motif close to the TTS. FSM and ISM transcripts were required to have support in both their 5′ and 3′ ends to pass the filter. For the rest of the transcript models, it was required that all splice junctions were supported by at least three short reads or were canonical

junctions. Additionally, transcript models were filtered out if labeled as an intrapriming artifact (60% of As in the 20 bp downstream of the reported TTS at the genomic level), if one or more splice junctions were flagged as generated by reverse transcriptase switching, or if they were mono-exonic.

*Machine learning filter.* The definition of true-positive and true-negative transcript model sets is critical for the performance of the machine learning filter. Given that TSS and TTS orthogonal data were available, the true-positive set was defined using FSM multi-exonic transcript models with CAGE-seq support at the 5′ end, Quant-seq support at the 3′ end, and canonical splice junction. Meanwhile, the true-negative set consisted of non-FSM multi-exonic transcripts lacking support in at least one of their ends or having a noncanonical splice junction. In total, 3,000 transcript models with these properties were randomly sampled for each set. Importantly, the attributes used to define true-positive and true-negative sets, that is, CAGE-seq and Quant-seq support and canonical status of the splice junction, were removed from the set of variables used for model training to prevent overfitting. To achieve this, classification column names were supplied to the –remove_columns option in the SQANTI3 machine learning filter.

**Running SQANTI3 rescue.** To perform the rescue step, the reference annotation was first characterized using SQANTI3 QC and the same additional data used during lrRNA-seq transcriptome QC (Supplementary Methods). For WTC11, the GENCODE v39 human transcriptome was used as both query and reference annotation. Of note, this additionally included SIRVs. Importantly, SQANTI3 ignores reference transcripts shorter than 200 bp by default to avoid spurious matches to noncoding genes and annotated fragments that should not be captured through lrRNA-Seq. Given that SQANTI3 ignores by default reference transcripts shorter than 200 bp to avoid spurious matches, to appropriately characterize the reference, we set –min_ref_len 0 in SQANTI3 QC to avoid the incorrect classification of short transcripts.

Next, SQANTI3 rescue was run using the SQANTI3 QC classification file from the reference and the SQANTI3 filter output obtained after filtering the long read-defined transcriptome as input. In addition, for post-machine learning filter rescue runs, we supplied the pre-trained random forest classifier used for transcriptome filtering. For post-rules filter rescue runs, the JSON file containing the rules and requisites used upon filtering was supplied as input to SQANTI3 rescue.

**SIRV evaluation metrics.** To assess the ability of SQANTI3 to accept or reject transcripts accurately, Lexogen SIRVs[33] introduced during library preparation were used. Given that SQANTI3 uses the reference annotation to categorize the transcript models and this affects the curation process, the annotation of the SIRVs was modified to include novel isoforms and isoforms annotated but not present in the sample. To achieve this, the annotation of the SIRV genes was added to the GENCODE human reference transcriptome (v39), including the insufficient and the over-annotated versions available at the Lexogen website https://www.lexogen.com/wp-content/uploads/2021/06/SIRV_Set4_Norm_Sequences_20210507.zip.

We considered an isoform as true positive if it matched a transcript model of the complete and correct annotation as an FSM reference match. Depending on the spike-in matched, it could be a known or novel true positive if it was present in the modified annotation or not, respectively. When the transcript model could be associated with the transcript present in the sample but differed by more than 50 bp in any of the ends, this was counted as a partial true positive. Transcripts matching a false SIRV present in the reference were computed as an over-annotation false positive. The rest of the transcripts that were classified as NIC or NNC compared with any annotation (complete or modified) were reported as false positives. Moreover, we measured the number of isoforms considered novel during the SQANTI3 curation

process, which includes novel true positive (TP) and false positive (FP). Using these figures, sensitivity, precision, F-score, false discovery rate (FDR), over-annotation detection rate (ODR) and novel detection rate (NDR) were calculated as follows:

$$\text{Sensitivity} = \frac{\text{known TP} + \text{novel TP}}{\text{no. of SIRVs introduced}}$$

$$\text{Precision} = \frac{\text{known TP} + \text{novel TP}}{\text{no. of SIRVs detected}}$$

$$\text{Fscore} = \frac{2 \times \text{Sensitivity} \times \text{Precision}}{\text{Sensitivity} + \text{Precision}}$$

$$\text{FDR} = \frac{\text{FP} + \text{Over} - \text{annot FP} + \text{Partial TP}}{\text{no. of SIRVs detected}}$$

$$\text{ODR} = \frac{\text{Over} - \text{annot FP}}{\text{no. of SIRVs detected}}$$

$$\text{NDR} = \frac{\text{novel TP} + \text{FP}}{\text{no. of SIRVs detected}}$$

### Reporting summary
Further information on research design is available in the Nature Portfolio Reporting Summary linked to this article.

## Data availability
Data used in this manuscript are publicly available under the following accession codes: for WTC11 analysis, lrRNA-seq data were retrieved from ENCODE ENCSR507JOF, srRNA-seq data were retrieved from ENCODE ENCSR673UKZ, CAGE-seq data were retrieved from GEO GSE185917, and Quant-seq data were retrieved from ENCODE ENCSR322MWL; for K562 analysis, ENCODE ENCSR917JIA experiment data were used, including long-read transcriptome (ENCFF584GRG), srRNA-seq (ENCSR792OIJ) and CAGE-seq data (ENCSR000CJN); for H1-endoderm analysis, data were retrieved from ENCODE, including lrRNA-seq data from accessions ENCSR271KEJ (H1-hESC) and ENCSR127HKN (H1-DE), srRNA-seq data from accessions ENCSR588EJX (H1-hESC) and ENCSR266XAJ (H1-DE), and Quant-seq data from accessions ENCSR198UNH (H1-hESC) and ENCSR198UNH (H1-DE). Reference data from PolyASite[32] and refTSS[31] were used to validate 3′ and 5′ ends. IsoAnnot annotation for human used the InterProScan (https://www.ebi.ac.uk/interpro/), UniProt (https://www.uniprot.org/) and Modi-DB (https://mobidb.org/) databases to functionally annotate reference transcripts. All of the files used to generate the results in this paper are publicly accessible at http://conesalab.org/SQANTI3/. For an easier exploration of WTC11 transcript models identified with IsoSeq3 and their characterization with SQANTI3, a public UCSC Genome Browser Track hub was generated (http://conesalab.org/SQANTI3/WTC11/SQANTI3_hub/hub.txt) including the orthogonal data used for validation. We also make the results of the differentiation of hESC H1 cells to endoderm available in a ready-to-use format on tappAS (http://conesalab.org/SQANTI3/H1_endo/tappAS_files/). Source data are provided with this paper.

## Code availability
The SQANTI3 software is available at https://github.com/ConesaLab/SQANTI3, with extensive documentation on the GitHub wiki site https://github.com/ConesaLab/SQANTI3/wiki. All results in this study were generated using SQANTI3 v5.1. Code used for generating the figures in the main text and in the Supplementary Notes is available at https://github.com/ConesaLab/SQANTI3.

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

## Acknowledgements

This work has been funded by NIH grant 7R21HG011280-02, by the Spanish Ministry of Science grants BES-2016-076994 and PID2020-119537RB-100, and by the Comunitat Valenciana grant ACIF/2018/290.

## Author contributions

F.J.P.-P. developed and implemented novel SQANTI3 QC features, developed and expanded the SQANTI3 rules filter, designed and performed data analyses, documented SQANTI3 QC, and wrote the manuscript. A.A.-L. refactored and improved the SQANTI3 machine learning filter, developed and implemented the SQANTI3 rescue module, designed the SQANTI3 documentation site, documented SQANTI3 filter and rescue, and wrote the manuscript. L.K. contributed to SQANTI3 quality control implementations. P.S. developed and implemented IsoAnnotLite. J.M.-T. contributed to data analysis. R.A. contributed to the novel SQANTI3 quality control graphical output. E.E.-M. contributed to SQANTI3 machine learning filter development. T.L. contributed to SQANTI3 quality control graphical output. A.N. contributed to rescue validation analyses. L.M. contributed to conceptualization. E.T. contributed to defining the SQANTI3 categories, developed usage of additional data in SQANTI3 quality control, and conceived the rules filter strategies. A.C. conceived and supervised the study. All authors read and agreed to the contents of the manuscript.

## Competing interests

The authors declare no competing interests

## Additional information

**Extended data** are available for this paper at https://doi.org/10.1038/s41592-024-02229-2.

**Correspondence and requests for materials** should be addressed to Ana Conesa.

**Reviewer Recognition** *Nature Methods* thanks the anonymous reviewer(s) for their contribution to the peer review of this work. Peer reviewer reports are available. Primary Handling Editor: Lei Tang, in collaboration with the *Nature Methods* team.

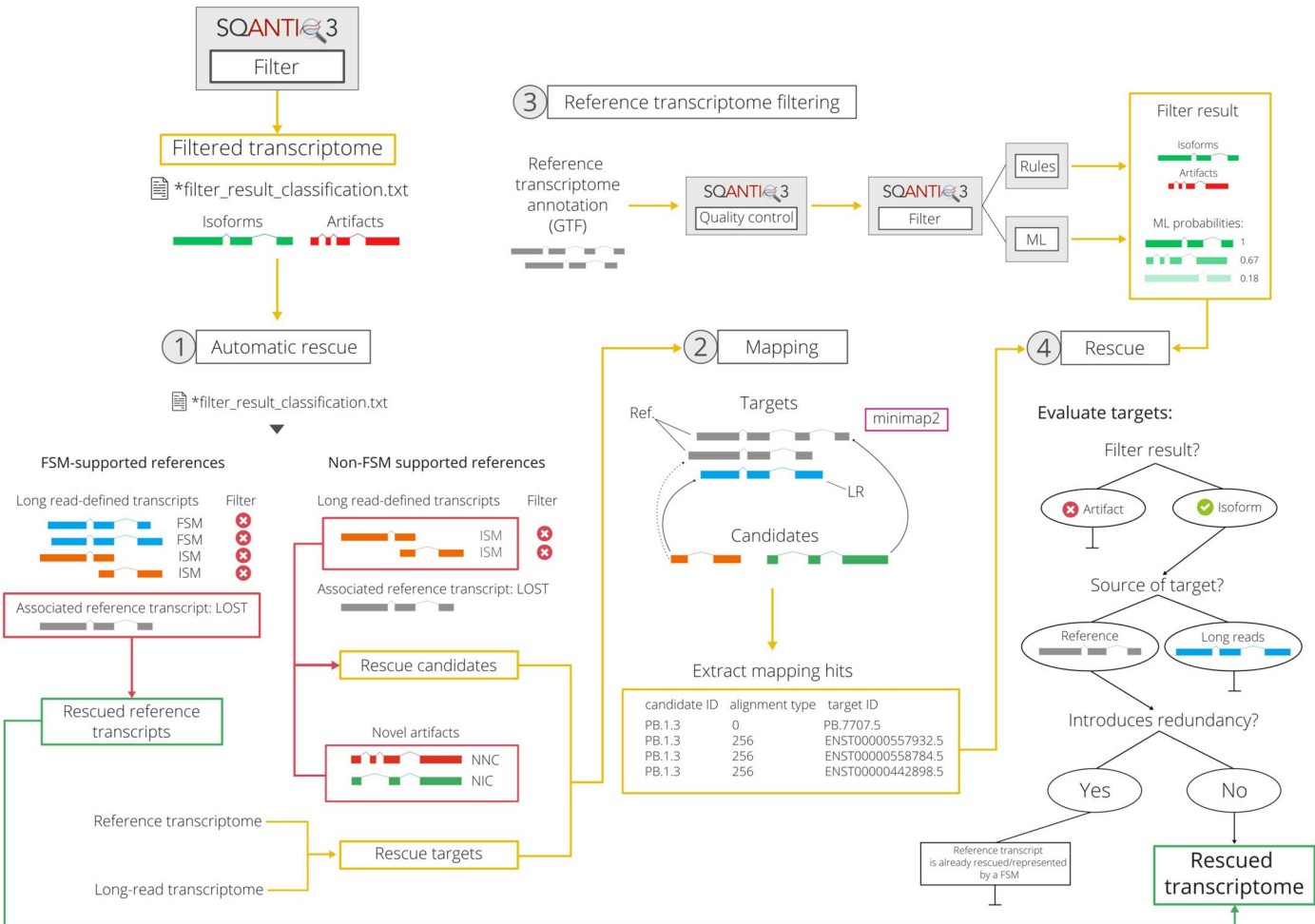

**Extended Data Fig. 1 | SQANTI3 Rescue workflow.** 1) If an FSM-supported reference transcript is lost during the filtering, the version of the reference is automatically rescued. 2) The rest of the LR-defined transcript models filtered out (rescue candidates) are mapped against the reference transcriptome combined with the accepted LR-defined isoforms (rescue targets), allowing several hits per candidate. 3) Reference transcriptome was previously evaluated and filtered with the same data and criteria as the LR-defined transcripts. 4) Rescue is completed by evaluating targets. They need to pass the filtering and not increase the redundancy, meaning that if the target is an LR-defined transcript present or it is a reference transcript already represented as an FSM in the filtered transcriptome, these targets will not be added to the final annotation. LR: Long-read, ML: Machine Learning, FSM: Full-Splice-Match, ISM: Incomplete-Splice-Match, NIC: Novel-In-Catalog, NNC: Novel-Not-In-Catalog.

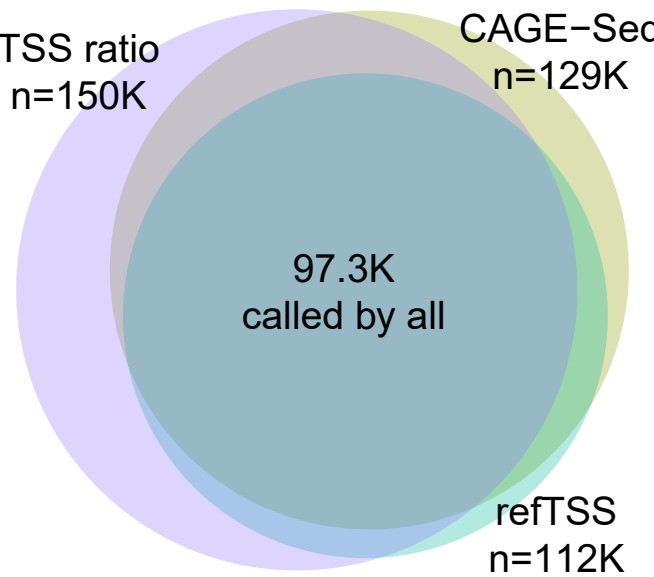

**Extended Data Fig. 2 | Agreement in TSS validation using different data sources. of additional information.** Number of TSS identified using the TSS ratio (threshold=1.5) based on matching short-reads RNA-seq data, sample-specific CAGE-seq data and the refTSS database. TSS: Transcript Starting Site.

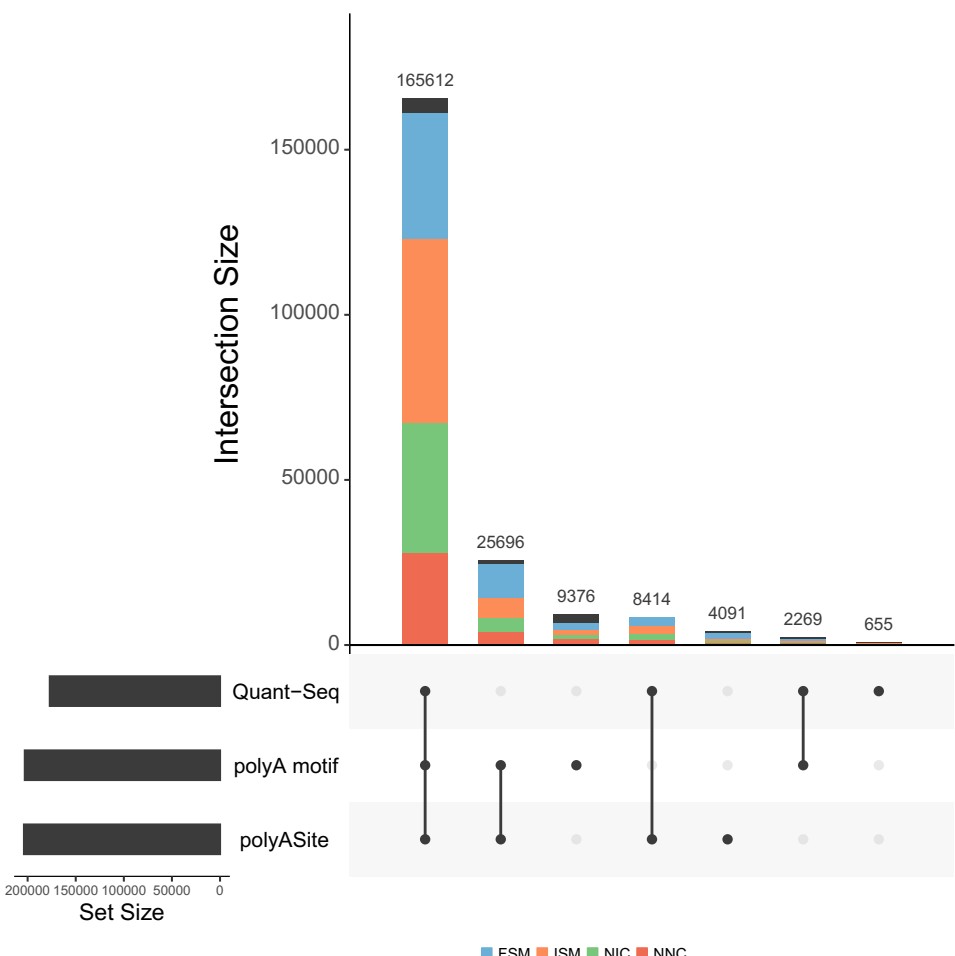

**Extended Data Fig. 3 | Agreement in TTS validation using different data sources.** Number of TTS identified using sample-specific Quant-seq data, presence of polyA motif and the PolyASite database. WTC11 PacBio lrRNA-seq data. TTS: Transcript Termination Site.

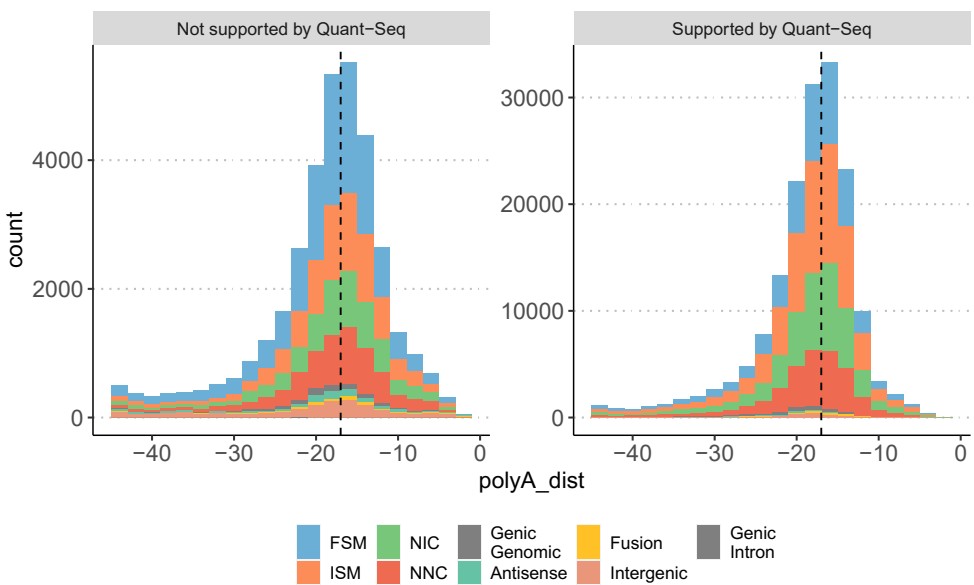

**Extended Data Fig. 4 | Frequency distribution of the transcript model distances between their detected polyA motif and the closest reference polyA site.** Data are stratified by SQANTI3 structural category and separated according to the existing Quant-seq data support. WTC11 PacBio lrRNA-seq data.

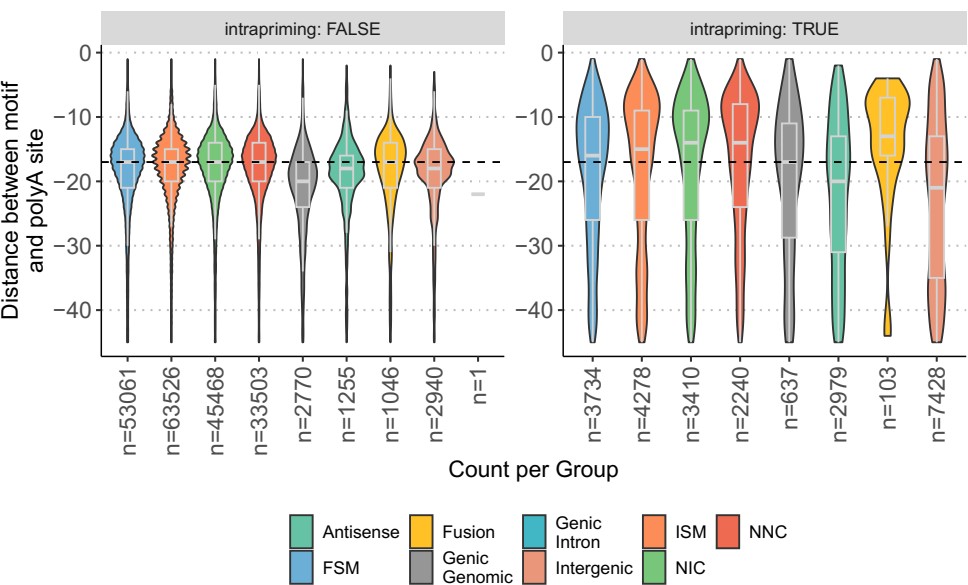

**Extended Data Fig. 5 | Distribution of transcript model distances between their detected polyA motif and the closest reference polyA site.** Data are broken-down by SQANTI3 structural category and separated depending on whether transcript models were flagged as potential intrapriming artifact. Boxes indicate median (middle line), 25th (Q1) and 75th (Q3) percentiles (box hinges); whiskers represent min = Q1 - 1.5 · Interquartile Range (IQR) and max = Q3 + 1.5 · IQR; dots constitute outliers. FSM: Full-Splice-Match. ISM: Incomplete-Splice-Match, NIC: Novel-In-Catalog, NNC: Novel-Not-In-Catalog.

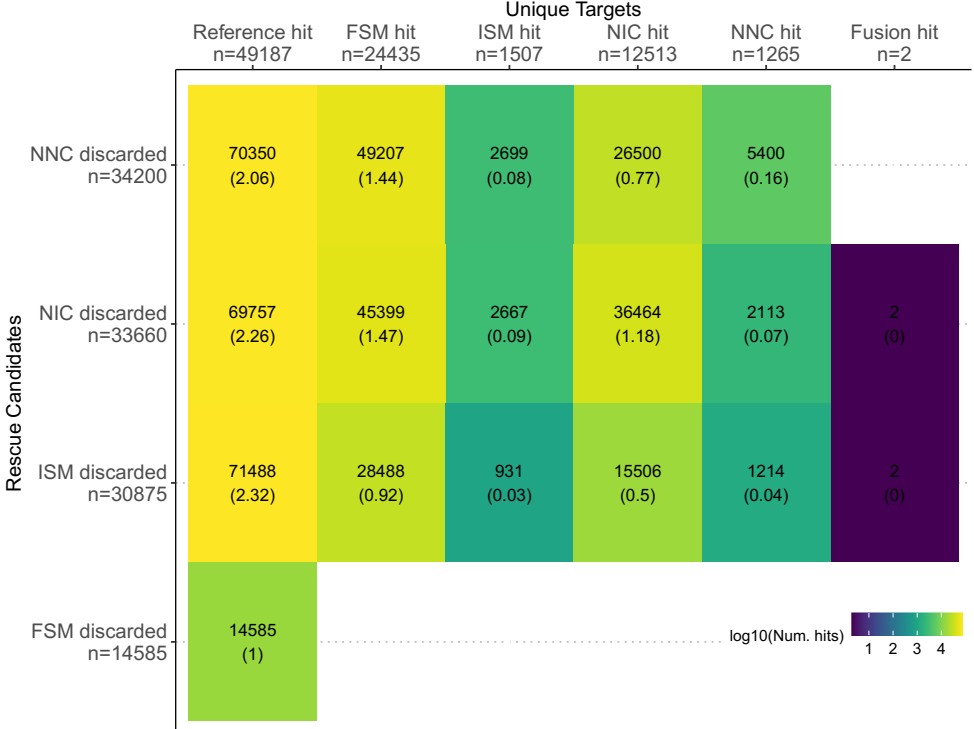

**Extended Data Fig. 6 | Relationship between the SQANTI3 structural categories of discarded transcripts (rescue candidates) and their rescue targets in the Machine Learning (ML) - High Input Sample filtering scenario.** Rescue candidates are shown in the y-axis, stratified by structural category. Candidates correspond to transcripts discarded by the ML filter, that is artifacts. Rescue targets are shown in the x-axis, spread across structural categories and including reference transcriptome hits. Targets correspond to transcripts mapped by artifacts during the rescue process. In this mapping process, each candidate can map to multiple targets, which are similar to the candidate in sequence and exon structure. Heatmap color therefore corresponds to the number of hits (log10) involving each possible pair of structural categories, indicating the amount of structural similarity among categories detected during rescue. Within the tiles, the total number of candidate target pairs is shown, including the mean number of hits per candidate for each category pair between parentheses. FSM candidates only match reference targets, since they are only considered for automatic rescue. WTC11 PacBio lrRNA-seq data. FSM: Full-Splice-Match, ISM: Incomplete-Splice-Match, NIC: Novel-In-Catalog, NNC: Novel-Not-In-Catalog.

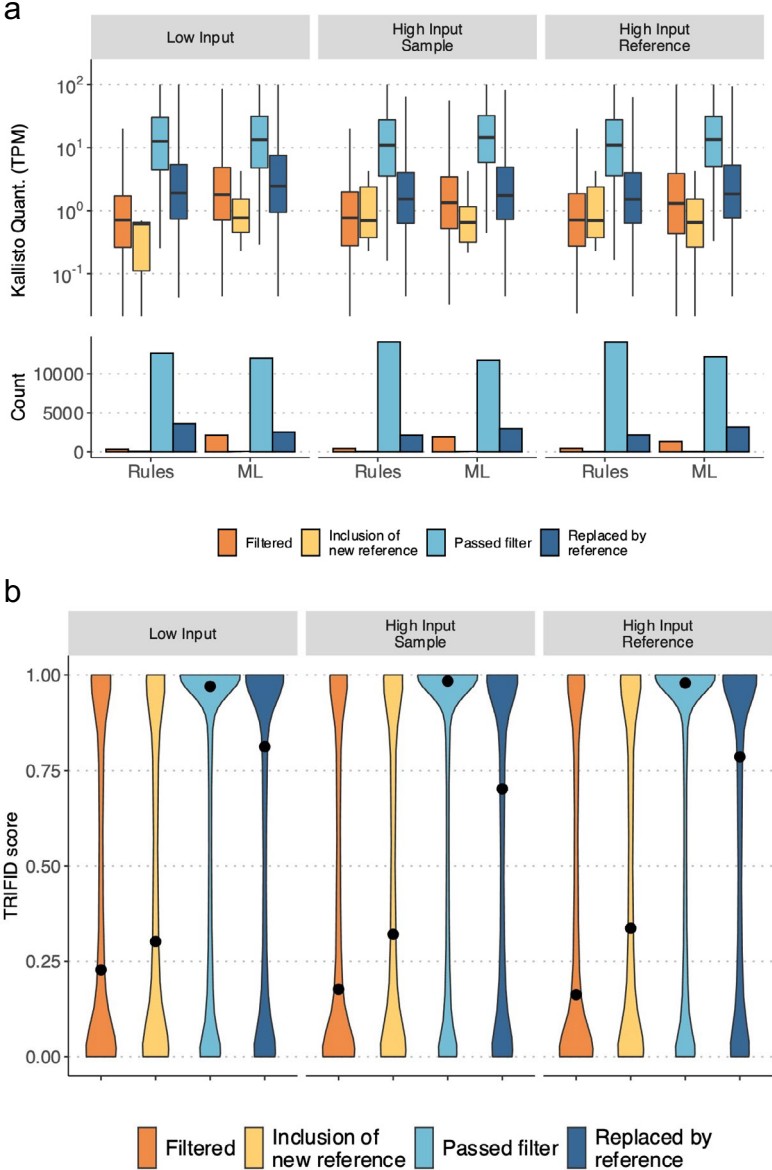

**Extended Data Fig. 7 | Expression and functional properties of rescued transcripts. a**, Distribution of expression values (TPM) of known transcripts detected as Full-Splice-Match or Incomplete-Splice-Match. **b**, TRIFID scores of known transcripts identified in each filtering and rescue scenario. Filtered transcripts (orange) did not pass the corresponding filter and were not eventually rescued. Transcripts filtered but recovered by introducing an isoform from the reference (dark blue) represent the rescue strategy's fundamental purpose. In exceptional cases, transcripts models not initially detected were included in the final transcriptome (yellow) via rescue.

# Reporting Summary

## Statistics

For all statistical analyses, confirm that the following items are present in the figure legend, table legend, main text, or Methods section.

| n/a | Confirmed | |
|---|---|---|
| ☐ | ☒ | The exact sample size (*n*) for each experimental group/condition, given as a discrete number and unit of measurement |
| ☐ | ☒ | A statement on whether measurements were taken from distinct samples or whether the same sample was measured repeatedly |
| ☐ | ☒ | The statistical test(s) used AND whether they are one- or two-sided<br>*Only common tests should be described solely by name; describe more complex techniques in the Methods section.* |
| ☐ | ☒ | A description of all covariates tested |
| ☐ | ☒ | A description of any assumptions or corrections, such as tests of normality and adjustment for multiple comparisons |
| ☐ | ☒ | A full description of the statistical parameters including central tendency (e.g. means) or other basic estimates (e.g. regression coefficient) AND variation (e.g. standard deviation) or associated estimates of uncertainty (e.g. confidence intervals) |
| ☐ | ☒ | For null hypothesis testing, the test statistic (e.g. *F*, *t*, *r*) with confidence intervals, effect sizes, degrees of freedom and *P* value noted<br>*Give P values as exact values whenever suitable.* |
| ☒ | ☐ | For Bayesian analysis, information on the choice of priors and Markov chain Monte Carlo settings |
| ☐ | ☒ | For hierarchical and complex designs, identification of the appropriate level for tests and full reporting of outcomes |
| ☒ | ☐ | Estimates of effect sizes (e.g. Cohen's *d*, Pearson's *r*), indicating how they were calculated |

*Our web collection on statistics for biologists contains articles on many of the points above.*

## Software and code

Policy information about availability of computer code

| Data collection | No software was used for data collection |
|---|---|
| Data analysis | https://github.com/ConesaLab/SQANTI3 |

For manuscripts utilizing custom algorithms or software that are central to the research but not yet described in published literature, software must be made available to editors and reviewers. We strongly encourage code deposition in a community repository (e.g. GitHub). See the Nature Portfolio guidelines for submitting code & software for further information.

## Data

Policy information about availability of data

All manuscripts must include a data availability statement. This statement should provide the following information, where applicable:
- Accession codes, unique identifiers, or web links for publicly available datasets
- A description of any restrictions on data availability
- For clinical datasets or third party data, please ensure that the statement adheres to our policy

Data used in this manuscript is publicly available under the following accession codes: for WTC11 analysis lrRNA-Seq (ENCODE ENCSR507JOF), srRNA-Seq (ENCODE ENCSR673UKZ), CAGE-Seq (GEO GSE185917) and Quant-Seq (ENCODE ENCSR322MWL); for K562 analysis it was used ENCODE ENCSR917JIA experiment data with its long-read-based transcriptome (ENCODE ENCFF584GRG), srRNA-Seq (ENCODE ENCSR792OIJ) and CAGE-Seq data (ENCODE ENCSR000CJN); and for H1-endoderm analysis data included lrRNA-Seq ENCODE ENCSR271KEJ (H1-hESC) and ENCSR127HKN (H1-DE), srRNA-Seq ENCODE ENCSR588EJX (H1-hESC) and ENCSR266XAJ (H1-

DE), and Quant-Seq ENCODE ENCSR198UNH (H1-hESC) and ENCSR198UNH (H1-DE). Reference data from databases such as polyASite and refTSS was used to validate 3' and 5'-ends. IsoAnnot annotation for human used InterProScan (https://www.ebi.ac.uk/interpro/), UniProt (https://www.uniprot.org/) and Modi-DB (https://mobidb.org/) databases to functionally annotate reference transcripts.

All the files used to generate the results in this paper are publicly accessible at http://conesalab.org/SQANTI3/. For an easier exploration of WTC11 transcript models identified with IsoSeq3 and their characterization with SQANTI3, a specific and public Track hub was generated for the UCSC Genome Browser (hub URL: http://conesalab.org/SQANTI3/WTC11/SQANTI3\_hub/hub.txt), including the orthogonal data used for validation. We make a special emphasis on the availability of the results of hESC H1 cells to endoderm differentiation in a ready-to-use format on tappAS http://conesalab.org/SQANTI3/H1_endo/tappAS_files/).

# Human research participants

Policy information about <u>studies involving human research participants and Sex and Gender in Research.</u>

| | |
|---|---|
| Reporting on sex and gender | NA |
| Population characteristics | NA |
| Recruitment | NA |
| Ethics oversight | NA |

Note that full information on the approval of the study protocol must also be provided in the manuscript.

# Field-specific reporting

Please select the one below that is the best fit for your research. If you are not sure, read the appropriate sections before making your selection.

☒ Life sciences    ☐ Behavioural & social sciences    ☐ Ecological, evolutionary & environmental sciences

For a reference copy of the document with all sections, see nature.com/documents/nr-reporting-summary-flat.pdf

# Life sciences study design

All studies must disclose on these points even when the disclosure is negative.

| | |
|---|---|
| Sample size | Three different samples (WTC11, K562 and H1-endoderm cells) were obtained from public data bases with their corresponding replicates. No statistical analysis involving the utilization of different cell lines was performed in the study and these 3 samples were used to illustrate wide applicability.  Statistical analyses evaluating association between transcript quality descriptors such as CAGE-peak coverage, polyA motif, etc, were performed on the whole transcriptome datasets with over 100,000 transcripts, that act as the number of observations or sample size for these test and is considered sufficient for the purposes of these tests. |
| Data exclusions | Data wasn't excluded in any case |
| Replication | Experimentation per se was not used in this study and the utilization of multiple public datasets was sufficient for the nature of the analyses performed. As indicated in the Sample Size section, the utilization of the whole transcriptome provided sufficient number of observations to support the statistical analyses. |
| Randomization | No randomization was done except for the selection of TP and TN sets. Transcript models were randomly selected following the criteria described in the paper for each data availability scenario simulated in order to run the ML-filtering algorithm. |
| Blinding | In general blinding was not applicable to the study. Predictions were performed without knowledge of the ground truth  to be able to compute performance metrics for the SQANTI3 curation strategy. |

# Reporting for specific materials, systems and methods

We require information from authors about some types of materials, experimental systems and methods used in many studies. Here, indicate whether each material, system or method listed is relevant to your study. If you are not sure if a list item applies to your research, read the appropriate section before selecting a response.

## Materials & experimental systems

| n/a | Involved in the study |
|-----|----------------------|
| ☒ | ☐ Antibodies |
| ☒ | ☐ Eukaryotic cell lines |
| ☒ | ☐ Palaeontology and archaeology |
| ☒ | ☐ Animals and other organisms |
| ☒ | ☐ Clinical data |
| ☒ | ☐ Dual use research of concern |

## Methods

| n/a | Involved in the study |
|-----|----------------------|
| ☒ | ☐ ChIP-seq |
| ☒ | ☐ Flow cytometry |
| ☒ | ☐ MRI-based neuroimaging |

