## [Peer Review File · Nature Methods]

Peer Review Information

Manuscript Title: SQANTI3: curation of long-read transcriptomes for accurate identification of known and novel isoforms

Corresponding author name(s): Ana Conesa

Editorial Notes:

Reviewer Comments & Decisions:

Decision Letter, initial version:

15th Aug 2023

Dear Dr Conesa,

Your Article, "SQANTI3: curation of long-read transcriptomes for accurate identification of known and novel isoforms", has now been seen by 3 reviewers. As you will see from their comments below, although the reviewers find your work of considerable potential interest, they have raised a number of concerns. We are interested in the possibility of publishing your paper in Nature Methods, but would like to consider your response to these concerns before we reach a final decision on publication.

All reviewers find the presentation of the manuscript can be improved and are concerned about the conceptual novelty presented in SQANTI3. We therefore invite you to shorten your manuscript to Brief Communications (~1500 words). As you will see in our Editorial, <https://www.nature.com/articles/s41592-020-0927-4> the Brief Communications format is designed to publish updated versions of existing software tools.

When revising your paper,

* in the main text, please include a brief introduction targeted for biologist end-users, a brief description of the algorithm, a description of the new features of SQANTI3, and presentation of the validation results and applications. You may have 2 figures in the main text and have the remaining figures in Extended Data and Supplementary Information. You can also have Supplementary Notes to describe results in more details.

- * include a point-by-point response to the reviewers and to any editorial suggestions
- * please underline/highlight areas with significant changes to facilitate review of the revised manuscript
- * address the points listed described below to conform to our open science requirements
- * ensure it complies with our general format requirements as set out in our guide to authors at www.nature.com/naturemethods
- * resubmit all the necessary files electronically by using the link below to access your home page

[REDACTED]

We hope to receive your revised paper within 8 weeks. If you cannot send it within this time, please let us know. In this event, we will still be happy to reconsider your paper at a later date so long as nothing similar has been accepted for publication at Nature Methods or published elsewhere.

OPEN SCIENCE REQUIREMENTS

REPORTING SUMMARY AND EDITORIAL POLICY CHECKLISTS

DATA AVAILABILITY

All novel DNA and RNA sequencing data, protein sequences, genetic polymorphisms, linked genotype and phenotype data, gene expression data, macromolecular structures, and proteomics data must be deposited in a publicly accessible database, and accession codes and associated hyperlinks must be provided in the "Data Availability" section.

CODE AVAILABILITY

Please include a "Code Availability" subsection in the Online Methods which details how your custom code is made available. Only in rare cases (where code is not central to the main conclusions of the paper) is the statement "available upon request" allowed (and reasons should be specified).

For more information on our code sharing policy and requirements, please see: <https://www.nature.com/nature-research/editorial-policies/reporting-standards#availability-of->

computer-code

ORCID

Sincerely,
Lei

Lei Tang, Ph.D.
Senior Editor
Nature Methods

Reviewers' Comments:

Reviewer #1:

Remarks to the Author:

This manuscript describes SQANTI3, an upgraded version of the well used and well supported SQANTI tool for assessing the quality of a transcriptome reconstruction from long read RNA/cDNA sequencing. The key upgrades in SQANTI3 are the inclusion of new transcript model subcategories, improved annotation of alternative 5' and 3' transcripts, a rules-based filter for discarding low quality transcripts, and a new program for rescuing discarded transcripts.

As a reader with a general understanding of both genome annotation and long read RNA sequencing, but not working directly on this problem, I found the presentation of the results quite hard to follow. The authors provide a commendable amount of data to demonstrate the use of SQANTI3 but the sheer amount of results, statistics, acronyms and interpretations are difficult to understand for this non-expert. I feel a simpler presentation, at reduced length, would greatly assist the reader in identifying the key messages to take away from this work. The current structure would probably be understandable and informative to those directly involved in this line of research but as this comes at the cost of a wider readership the authors should think carefully about who this manuscript is targeted towards. The discussion section is at a higher level and therefore more understandable. As are the methods, which are mostly intuitive heuristics.

I feel a more detailed comparison between SQANTI3 and SQANTI, or any competing software packages, would help make the advantages of the new features more apparent. For example all of the filtering results in Figure 3 are presented without comparison to other methods (like SQANTI), making their interpretation difficult. The lack of true ground truth data for WTC11 (and the other samples sequenced) also makes it difficult to assess the results as the arguments mainly rely on supporting short read datasets which have their own biases (e.g. possible complications of mapping to repetitive regions)

The quality of the software itself is generally high, with very good documentation (install instructions, descriptions of output) and usage examples. The authors should be commended on all of these points as it enhances the usability of this work.

One minor note, "differentially expressed" or "differential expression" is used in a few places instead of "differentially/differential"

Reviewer #2:

Remarks to the Author:

Overall summary:

Pardo-Palacios and Arzalluz-Luque et al present a much enhanced version (SQANTI3) of their previously published SQANTI software. SQANTI3 now has enhanced capacity for describing the 3' and 5' ends of transcripts. It extends its previous artifact removal strategy, in order to remove artifacts either using a rules-based or a machine learning-based filter. Moreover, it has added a rescue function that allows reads that are viewed as artifacts to be assigned to the most likely transcript. Last but not least, it also incorporates IsoAnnotLite, a previous tool by the same lab, which allows isoform interpretation.

SQANTI already possesses a large user base and most likely SQANTI3 will equally be widely used. Nevertheless, there are some aspects of the manuscript, which should be addressed.

Major comments:

1. At times, the manuscript is written in a manner that almost requires the reader to have read the previous SQANTI. The authors should take great care in going over the manuscript and asking if they can make sections more independent.

2. Regarding 3' end and 5' end validation: The authors employ an extremely low pseudocount, which can lead to artifacts: One read downstream of the TSS and none upstream would lead to a ratio of 101 (1.01/0.01), while 5 reads downstream and one read upstream (a much more informative situation) would lead to a ratio of ~5. This is exacerbated by using the maximum from distinct replicates, which means that one bad replicate can dominate the score. Unless this is already implemented, the authors should implement justified cutoffs of upstream + downstream reads (for example ≥ 10) and replace the maximum by a media (or 3rd quantile) over replicates. Moreover, they refer to a cutoff value of 1.5 ... which seems very low.

3. regarding the machine learning-based filter (section 4.1.2 and main text)

3a) The balancing of TP and TN datasets is an advantage and the sample-size flexibility makes it more user friendly. However, the danger is that users will go too low and therefore produce wrong results. The authors need to investigate how performance is determined by sample size and not allow the user to below the smallest size at which performance can still be guaranteed.

3b) Likewise for the definition of the TN set a probability cutoff of 0.7 is used. This suggests that this set is not pure at all. Especially, if we think of this as analogous to a multiple testing setting. The authors should empirically define the best cutoff.

Again, some users may appreciate the ability to redefine this cutoff. However, if they do so without any guidance on the implications, very wrong results could arise, which would be a disservice to the field.

3c) regarding "Specifically, HIS-ML was found to yield the most stringent filtering (173, 864 potential artifacts), while HIS-Rules was the most lenient (88,786 potential artifacts)". Detailed investigations of what the ML filter picks up should enable the definition of a better rules-based filter.

3d) Overall, section "2.3 Characterization of the SQANTI3 Filter" shows drastic differences between HIS-ML and HIS-Rules: "Specifically, HIS-ML was found to yield the most stringent filtering (173,864 potential artifacts), while HIS-Rules was the most lenient (88,786 potential artifacts)." This section should end with a very strong statement on which one is to be used under which circumstances. As is, users will choose the one that better fits their expectations, which invites false results.

4) regarding the rescue mode

Overall, the idea of a rescue module is laudable. In my mind the use case is especially genes that were lost. For example, if a downstream application is to quantify gene expression (whether with long or short reads), losing an entire gene is problematic. When it comes to isoforms, however, rescuing an isoform should be only done, when one is close-to-certain that it is correct. This can be accepted for FSM-based rescues. For ISM-based however, correctness seems questionable. There are two possible ways around this:

(i) perform experimental validations on ISM, NIC, and NNC -based rescues: take 10 randomly and validate by RT-PCR or similar.

(ii) Alternatively, limit the rescue to very-high-confidence rescues (that would probably mean FSM-based), for which problems are less likely to arise.

5) regarding the direct RNA transcriptome

It seems this section is meant to show the versatility of SQANTI3. However, the implementation of a new "tailored" (third) filter as well as an "ad-hoc" rescue criterion may also motivate readers to create their own approaches. If well implemented, this may be useful, but most users will lack the detailed knowledge to guarantee careful implementation. Understandably, authors generally tend to dislike removing analysis that they have done – but I believe the paper (and the field!) would be best served by removing this section.

Additionally, rather than trying to "get something out of" any available dataset, SQANTI3 should make it very clear to users, if their data is not high-quality.

6) regarding the abstract

The authors should clarify what is novel in SQANTI3 with respect to previous versions. For example, it seems to me that "SQANTI3 provides an extensive naming framework to describe transcript model diversity in comparison to the reference transcriptome" equally applies to SQANTI.

Minor comments:

- Some parts of the manuscript are written in unconventional style. As an example: "Reference transcripts with FSM-based evidence for which all associated isoforms have been removed are recovered, i.e. automatic rescue." This should be reviewed.
- It appears that words like "transcripts" and "isoforms" are used in a rather synonymous way, although there does seem to be some distinction. It would be good to use language that is clear. For example, is the phrase "Reference transcripts with FSM-based evidence for which all associated isoforms have been removed" identical in meaning to "Reference genes with FSM-based evidence for which all reference transcript models have been removed"? If yes, the latter would avoid misinterpretations.
- Lines 247-257 would benefit from repeating the total size of the FSM and ISM numbers. Otherwise, counts in this paragraph are difficult to interpret.
- "The detection of a polyA motif was found to be significantly associated with 3' end validation by Quant-seq (p-value = 2.2×10^{-16} , Fisher's Exact Test, see Methods and Supplementary Fig.A2a)." The figure shows three distinct variables, but the phrase concentrates on two. The authors should explore different depictions focusing on the two variables mentioned in the text.
- There are some spelling mistakes, including "additionaldata" (with a missing space) in line 344.
- It is unclear what the "OR configuration" is in line 355
- The language sometimes lets readers guess what is exactly meant. For example, lines 205 and 446 use the term "automatic recues", but in between the term is not used. In line 446, the reader can struggle. Using "FSM-based rescue" or similar would enhance readability.
- In the rescue section, I first interpreted the phrase "10,389 isoforms and 2,884 genes were recovered from the reference transcriptome, while 36,811 transcripts and 2,638 known genes were included on average when the rescue was applied to the rules-filtered datasets" as that all isoforms came from the rescued genes. However, this would be a disturbingly high ratio in the latter case (36,811 transcripts for 2,638 known genes). I now think that isoforms were rescued from all genes. Either way, the language should make it very clear.
- The paragraph in lines 479-491 is difficult to read because of the very high abundance of abbreviations.
- In the direct RNA section, the authors mention "51,843 known genes". Given ~20k known protein-coding genes, this phrase begs for some explanations about lncRNAs their expression levels, etc (Most likely this is set is dominated by lncRNAs, which in most cases have rather few reads).

Reviewer #3:

Remarks to the Author:

Pardo-Palacios and Arzalluz-Luque present SQUANTI3 software for curating of RNA isoforms using long read data. They show performance and benchmarking analyses using long read direct RNA and cDNA datasets.

This software aims to fill the gap of well-developed tools for isoform curation using long read DRS and cDNA datasets. I like that the authors incorporated orthogonal markers in this process vs. focusing on just assembling/curating sequences together.

I have a few comments:

1. I appreciate the included details of the workflow. I would recommend also including a simpler description in addition for a non-RNA-Seq experts.
2. For the TSS/TTS variability from IsoSeq3 vs SQUANTI3, did the authors validate any of these sites experimentally or computationally? That could add compelling evidence.
3. How does the SQUANTI3 account for the 5'-terminal missing bases in nanopore RNA sequencing?
4. Similar to 3, is there a risk in curating a transcript where the 5'-end is not unambiguously determined. I.e., what is the measure of a true 5'-cap? Is it only the orthogonal markers?
5. How does SQUANTI3 handle retained-introns in transcripts?
6. The figures could be simplified, especially they feel dense and hard to read.
7. It would be useful to include the tutorial in GitHub, along with a couple of Jupyter notebooks.

Best wishes to the authors.

Author Rebuttal to Initial comments

Dear Dr. Lei

Thank you for giving us the opportunity to revise our manuscript "SQANTI3: curation of long-read transcriptomes for accurate identification of known and novel isoforms". Please, find below our point-by-point responses to both editorial and reviewers remarks. The length of the manuscript has been substantially reduced and we have put extra effort in providing a clear overview of the software's functionalities while highlighting differences with previous version. I hope you find the revised manuscript suitable for publication in Nature Methods.

Best regards

Ana Conesa, on behalf of the authors.

Editorial comments

We therefore invite you to shorten your manuscript to Brief Communications (~1500 words). As you will see in our Editorial, <https://www.nature.com/articles/s41592-020-0927-4> the Brief Communications format is designed to publish updated versions of existing software tools.

When revising your paper,

* in the main text, please include a brief introduction targeted for biologist end-users, a brief description of the algorithm, a description of the new features of SQANTI3, and presentation of the validation results and applications. You may have 2 figures in the main text and have the remaining figures in Extended Data and Supplementary Information. You can also have Supplementary Notes to describe results in more details.

We have made a substantial re-writing of the manuscript to adapt to the Brief Communication format. The main text describes the SQANTI3 tool, concept and main analysis workflow taking the biologist end-user reader in mind. We illustrate application to long-read transcriptome QC analysis and present some of the validation results. We have included 5 Supplementary Notes to provide extended information on features description, utilization of the QC options, validation, application to different data types and utilization for downstream analysis. We hope that with this changes the paper is more readable while still containing detailed information in the Supplementary Notes.

* include a point-by-point response to the reviewers and to any editorial suggestions

Point-by-point responses are provided below.

* please underline/highlight areas with significant changes to facilitate review of the revised manuscript

We have chosen not underlined the changes as the paper is basically re-written to adapt to the very reduced paper length. However, in the response to the reviewers we have indicated the line or lines in the revised manuscript when the point of revision is addressed.

* address the points listed described below to conform to our open science requirements

Done. Open science requirements are met.

* ensure it complies with our general format requirements as set out in our guide to authors at

www.nature.com/naturemethods

We have revised guidelines and understand the format requirements are met.

* resubmit all the necessary files electronically by using the link below to access your home page

Reviewer #1:

Remarks to the Author:

This manuscript describes SQANTI3, an upgraded version of the well used and well supported SQANTI tool for assessing the quality of a transcriptome reconstruction from long read RNA/cDNA sequencing. The key upgrades in SQANTI3 are the inclusion of new transcript model subcategories, improved annotation of alternative 5' and 3' transcripts, a rules-based filter for discarding low quality transcripts, and a new program for rescuing discarded transcripts.

As a reader with a general understanding of both genome annotation and long read RNA sequencing, but not working directly on this problem, I found the presentation of the results quite hard to follow. The authors provide a commendable amount of data to demonstrate the use of SQANTI3 but the sheer amount of results, statistics, acronyms and interpretations are difficult to understand for this non-expert. I feel a simpler presentation, at reduced length, would greatly assist the reader in identifying the key messages to take away from this work. The current structure would probably be understandable and informative to those directly involved in this line of research but as this comes at the cost of a wider readership the authors should think carefully about who this manuscript is targeted towards. The discussion section is at a higher level and therefore more understandable. As are the methods, which are mostly intuitive heuristics.

Thank you for your comments, which are in agreement with the editorial requests. We have substantially reduced the length and complexity of the manuscript to be drafted as a Brief Communication. Basically, the last two sections of the manuscript (analysis of a Nanopore dataset and Iso-Functional analysis) have been moved to Supplementary Notes and the description of the tool together with some validation aspects have been combined in one section. Other Supplementary Notes include the detailed comparison of SQANTI versions, the in-depth analysis of the QC utilities in SQANTI3 and the in-depth analysis of the filtering options. The manuscript includes now only two main figures. We hope this simplified version of the manuscript is more readable.

I feel a more detailed comparison between SQANTI3 and SQANTI, or any competing software packages, would help make the advantages of the new features more apparent. For example all of the filtering results in Figure 3 are presented without comparison to other methods (like SQANTI), making their interpretation difficult. The lack of true ground truth data for WTC11 (and the other samples sequenced) also makes it difficult to assess the results as the arguments mainly rely on supporting short read datasets which have their own biases (e.g. possible complications of mapping to repetitive regions)

Thank you for your remark. As far as we are aware, there are no other tools specifically for the quality control of long-reads transcript models. As for the comparison with SQANTI, the SQANTI settings will actually be those we have denoted as low input, as they are not using the TTS/TSS supporting data, which is one of the major novelties of SQANTI3. We have indicated in the manuscript that the low-input scenario equals SQANTI analysis (line 211). Please, note that due to the shortening of the paper to fit the Brief Communication format, this comparison of filtering scenarios is now presented in Supplementary Note 5. Moreover, although it is true that the WTC11 does not have a ground-truth, we did use the spike-in SIRVs dataset of the LRGASP project to provide an estimate of the accuracy of both the filter and the rescue steps in SQANTI3. These results are provided in figure 2g.

The quality of the software itself is generally high, with very good documentation (install instructions, descriptions of output) and usage examples. The authors should be commended on all of these points as it enhances the usability of this work.

Thank you very much for your compliments

One minor note, “differentially expressed” or “differential expression” is used in a few places instead of “differentially/differential”

Thank you for spotting this typo, which has been corrected.

Reviewer #2:

Remarks to the Author:

Overall summary:

Pardo-Palacios and Arzalluz-Luque et al present a much enhanced version (SQANTI3) of their previously published SQANTI software. SQANTI3 now has enhanced capacity for describing the 3' and 5' ends of transcripts. It extends its previous artifact removal strategy, in order to remove artifacts either using a rules-based or a machine learning-based filter. Moreover, it has added a rescue function that allows reads that are viewed as artifacts to be assigned to the most likely transcript. Last but not least, it also incorporates IsoAnnotLite, a previous tool by the same lab, which allows isoform interpretation.

SQANTI already possesses a large user base and most likely SQANTI3 will equally be widely used.

Nevertheless, there are some aspects of the manuscript, which should be addressed.

Major comments:

1. At times, the manuscript is written in a manner that almost requires the reader to have read the previous SQANTI. The authors should take great care in going over the manuscript and asking if they can make sections more independent.

Thank you for your remark. We understand that, as a updated version of an existing software, the manuscript should balance between comprehensiveness and the novelty aspects of the work. Following the editorial recommendations we have substantially reduced the manuscript to fit the Brief Communication format to make it more readable and consistent. Starting on line 72 we include a paragraph that provides a high-level description of SQANTI3 functionalities and the analysis workflow is presented in figure 1.a. To better explain the differences with previous SQANTI versions, former Supplementary Table 1 has been expanded and included in Supplementary Note 1 with a textual description of the overall SQANTI3 functionalities. We hope in this way the reader can have both a general overview of SQANTI3 functionalities (in the main text) as sufficient background information on previous SQANTI versions (in Supplementary Note 1) to fully understand the working of the tool.

2. Regarding 3' end and 5' end validation:

- The authors employ an extremely low pseudocount, which can lead to artifacts: One read downstream of the TSS and none upstream would lead to a ratio of 101 (1.01/0.01), while 5 reads downstream and one read upstream (a much more informative situation) would lead to a ratio of ~5. This is exacerbated by using the maximum from distinct replicates, which means that one bad replicate can dominate the score. Unless this is already implemented, the authors should implement justified cutoffs of upstream + downstream reads (for example ≥ 10) and replace the maximum by a media (or 3rd quantile) over replicates.

Thank you for pointing out this potential issue with the ratio TSS. We agree that a minimum coverage for this metric should make results more robust. Following the reviewer's suggestions, we have included now a mean coverage of 3x in the 100 bps downstream the TSS. We have chosen a mean coverage value rather than the number of reads for this threshold because TSS considers the mean coverage when computing the ratio. Additionally, the 3x value is in keeping with the 3x coverage value required for Illumina junction support. We re-run SQANTI3 with this new thresholds to generate all figures in the paper referring to TSS ratio. Moreover, we have now added a new option in SQANTI3 QC, `--ratio_TSS_metric`, that can take the values "max", "mean", "median" or "3quantile". The users can choose which metric suits best their analysis goals. For example, if they wish to discover aberrant isoforms, then max is probably a best option, while for building a reference transcriptome median or 3quantile is more appropriate. We have updated the documentation to explain when each option is best used.

- Moreover, they refer to a cutoff value of 1.5 ... which seems very low.

The value 1.5 was picked to account for possible nested TSS, or alternative TSS. If a non-dominant nested TSS is being actively expressed, we expect to see at least 50% increase in the short-read coverage of the first basepairs after the TSS defined by the single-molecule RNA-sequencing. However, this value can be empirically estimated from the distribution in the data. This distribution of TSS ratio values is now included in the SQANTI3 QC report to allow users to choose a good threshold. For the analyses presented in the paper, we took the 1.5 value as this distribution analysis indicated good agreement with CAGE support. Figures 1b and 1c include graphs that show that this threshold is statistically associated to CAGE support and is informative to discriminate between bona-fide TSS and artifacts. Moreover, when choosing filtering options, users have the possibility to use the cutoff value according to its distribution in case of applying a Rules filtering. If the Machine Learning option is employed, the classifier will decide the best discriminating threshold.

3. regarding the machine learning-based filter (section 4.1.2 and main text)

3a) The balancing of TP and TN datasets is an advantage and the sample-size flexibility makes it more user friendly. However, the danger is that users will go too low and therefore produce wrong results.

The authors need to investigate how performance is determined by sample size and not allow the user to below the smallest size at which performance can still be guaranteed.

We appreciate the reviewer's concern and, to provide clarification as to how TP/TN set sizes affects ML filter performance, we show a benchmark using set sizes ranging from 50 to 10K isoforms (see Figure 1 below) for the ML-HIS configuration, in which we report accuracy, sensitivity, precision and specificity as computed on the test set after training the random forest model. As a rule, we observe that performance tends to become unstable below 250 isoforms, and have therefore adjusted the minimum TP/TN set size in the tool accordingly, preventing users from running the ML filter under training sets smaller than 250 isoforms. In addition, we notice that ML performance parameters become stable with sizes larger than 1K isoforms, with only slight changes. We therefore consider that the decision to have 3K isoforms as the default set size in the SQANTI3 ML Filter is robust, and in agreement with these analysis results.

Figure 1: accuracy, precision, sensitivity and specificity for different TP and TN set sizes after running the SQANTI3 ML filter.

3b) Likewise for the definition of the TN set a probability cutoff of 0.7 is used. This suggests that this set is not pure at all. Especially, if we think of this as analogous to a multiple testing setting. The authors should empirically define the best cutoff. Again, some users may appreciate the ability to redefine this cutoff. However, if they do so without any guidance on the implications, very wrong results could arise, which would be a disservice to the field.

While the probability cutoff may be highly dataset-dependent, we established a default value of 0.7 for artifact calling in order to be additionally stringent and achieve a good balance between recall and precision, however, it should be noted that users can, after data exploration, flag artifacts according to their own threshold, or re-run the filter using new parameters. To facilitate user-level exploration of performance, and thanks to the reviewer's observations, we have expanded the ML filter report that is generated as output to include the positive (isoform) probability distribution generated after applying the random forest classifier to the data, as well as a table including all performance parameters and the confusion matrix generated on the test set.

In the case of the different ML data configurations, the probability distributions that were obtained are supplied in Figure 2 below. As it turns out, using a threshold of 0.7 captures the isoforms that match the high probability peak for all configurations, even when the bimodal distribution is less pronounced, as is the case for the HIS and HIR scenarios.

Figure 2: probability distributions after running the ML classifier for all input data configurations.

3c) regarding “Specifically, HIS-ML was found to yield the most stringent filtering (173, 864 potential artifacts), while HIS-Rules was the most lenient (88,786 potential artifacts)”. Detailed investigations of what the ML filter picks up should enable the definition of a better rules-based filter.

We agree with the reviewer that understanding the variables and thresholds used by the Machine Learning classifier is important and actually, the SQANTI3 Filter report includes a graph that indicates the importance of each parameter for the classifier and the value distributions between transcripts called isoforms and artifacts. In the manuscript we include the SQANTI3 filter report ML variable importance values in Figure 2d where users can appreciate which parameters had best discriminative power, and Figure 2e shows the distribution of each top importance variable per structural category to reveal different distributions between isoforms and artifact for each structural category. This highlights the complex nature of the ML filter, as expected from a machine learning classifier. Users may want to use this information to set hard threshold to design a Rules filtering strategy, as suggested by the reviewers. However, taking isolated variables to compose a user-defined Rules filtering may be difficult to configure. In fact, we show in figure 2g that the ML approach delivers better performance than a set of Rules designed for the same data.

3d) Overall, section “2.3 Characterization of the SQANTI3 Filter” shows drastic differences between HIS-ML and HIS-Rules: “Specifically, HIS-ML was found to yield the most stringent filtering (173,864 potential artifacts), while HIS-Rules was the most

lenient (88,786 potential artifacts).” This section should end with a very strong statement on which one is to be used under which circumstances. As is, users will choose the one that better fits their expectations, which invites false results.

Thank you for this important comment. We envision SQANTI3 as a QC and data assessment tool that can be used in different scenarios where long-read transcriptome sequencing is used for different purposes, as this will determine the strength of the stringency that should be applied. For example, for genome annotation, you may want to only consider high-confidence transcript models that are extensively supported by orthogonal as you may not want to over-populate reference transcriptomes with hundreds of thousands transcripts without enough evidence. However, in the context of cancer-derived rare isoform detection, you may want to be more lenient for discovery, especially if some downstream validation experiments are possible. SQANTI3 provide means to adapt your curation to different scenarios. We have added to the discussion a few sentences providing this type of recommendations (line 266): “For genome annotation, we recommend using extensively orthogonal data and applying ML-based filtering to obtain a set of high-confidence transcript models. In other applications that seek to detect rare novel transcripts, more lenient filtering may be applied to allow for discovery, especially when follow-up validations are planned. Finally, for isoform-resolved differential expression studies, filtering based on consistent detection across samples is advisable”.

4) regarding the rescue mode

Overall, the idea of a rescue module is laudable. In my mind the use case is especially genes that were lost. For example, if a downstream application is to quantify gene expression (whether with long or short reads), losing an entire gene is problematic. When it comes to isoforms, however, rescuing an isoform should be only done, when one is close-to-certain that it is correct. This can be accepted for FSM-based rescues. For ISM-based however, correctness seems questionable. There are two possible ways around this:

- (i) perform experimental validations on ISM, NIC, and NNC -based rescues: take 10 randomly and validate by RT-PCR or similar.
- (ii) Alternatively, limit the rescue to very-high-confidence rescues (that would probably mean FSM-based), for which problems are less likely to arise.

We thank the reviewer for acknowledging the value of the rescue step and we agree with that this step should be performed with confidence. To ensure the reliability of the rescue process, we SQANTI3 enforces that the same filtering criteria (either rules or ML-based) that was applied to long-read data holds also for the reference transcripts that are rescued. By doing this, we make sure that retrieved transcripts from the reference have the sample specificity and support by orthogonal data sources (CAGE, short-reads, etc.) required by the user and present in the filter output. This ensures consistency in the quality of the resulting transcriptome annotation. Nevertheless, to ensure the very-high-confidence rescues, we have now modified the SQANTI3 Rescue module by adding the `–mode` argument, for which ‘automatic’ is now the default. This mode only performs rescue of FSM isoforms to prevent the loss of genes and reference isoforms for which all long read counterparts were filtered out. Consequently, only in cases where users wish to perform more careful inspection of the rescue results, they may run SQANTI3 Rescue using `–mode ‘full’`. Note that, in this case, the tool generates a rescue table in which all potential rescue transcripts associated to each artifact are reported, including a detailed account of whether or not they were rescued and why, which allows users to make a critical assessment of the rescue results.

Regarding the first suggestion, performing additional PCR validation of the filtered output is out of the scope of this work. However, we leverage the LRGASP experimental validation data for the WTC11 data to assess our rescue strategy. In this project, a set of 45 PCR-validated reference transcript models were considered to assess if true transcripts can be actually rescued. Since the Rescue module was run in –mode “full”, any initially detected known transcript was recovered through automatic rescue. However, additional reference transcripts could be added to the transcriptome via mapping of ISM, NIC and NNC artifacts. To validate the usefulness of this mapping approach, we can compare how many known transcripts were detected by IsoSeq3 before SQANTI3 curation pipeline and if true reference transcripts not initially detected by IsoSeq3 were rescued.

Prior to Filter+Rescue, IsoSeq3 detected as FSM Reference Match 35 out of 45 validated isoforms. However, using the Rules+Rescue approach, 2 PCR-validated reference transcripts (not detected initially by IsoSeq3) were rescued from the reference via mapping of ISM, NIC and NNC discarded transcripts. It’s ~5% increase in sensitivity without any drawback of losing reliable TP transcripts.

5) regarding the direct RNA transcriptome

It seems this section is meant to show the versatility of SQANTI3. However, the implementation of a new “tailored” (third) filter as well as an “ad-hoc” rescue criterion may also motivate readers to create their own approaches. If well implemented, this may be useful, but most users will lack the detailed knowledge to guarantee careful implementation. Understandably, authors generally tend to dislike removing analysis that they have done – but I believe the paper (and the field!) would be best served by removing this section.

Additionally, rather than trying to “get something out of” any available dataset, SQANTI3 should make it very clear to users, if their data is not high-quality.

Since SQANTI3 is a versatile QC tool, we believe that is important to provide several examples of how different types of data may look like when performing the analysis. Also, we believe that using application to different type of data stresses the wide applicability of the tool. Finally, QC stringency criteria may be different depending on the analysis goal and we believe that the user should have the ultimate word on their curation process. These considerations led us to include the dRNA/TALON analysis section in the paper. However, due to the reformatting applied, we can now removed this section from the paper as requested and put it as a Supplementary Note, since we think it might be still useful for users working on alternative settings. We understand the concern of the reviewer for potential misuse of the tool and although this is something that cannot be 100% prevented for analysis frameworks with parameters chosen by the user, we have included a number of recommendations in the last paragraph of the paper (see above).

6) regarding the abstract

The authors should clarify what is novel in SQANTI3 with respect to previous versions. For example, it seems to me that “SQANTI3 provides an extensive naming framework to describe transcript model diversity in comparison to the reference transcriptome” equally applies to SQANTI.

In other to comply with editor and reviewer 2 requests, we are now following the recommendations by the editor to provide a summary description of the tool as a whole at the beginning of the manuscript (lines 72 to 83) and include the Supplementary Note 1 where we provide an extensive comparison of SQANTI3 functionality with previous versions.

Minor comments:

- Some parts of the manuscript are written in unconventional style. As an example: "Reference transcripts with FSM-based evidence for which all associated isoforms have been removed are recovered, i.e. automatic rescue." This should be reviewed.

This sentence have been removed and the message has been simplified in the new version. The roughly equivalent text in line 123 states "SQANTI3 includes a third Rescue module where artifacts are assigned to the most suitable reference or long-read transcript model by applying a two-step process that recovers reference transcript for discarded FSM, and orthogonal- data-supported alternatives for ISM, NIC and NNC"

- It appears that words like "transcripts" and "isoforms" are used in a rather synonymous way, although there does seem to be some distinction. It would be good to use language that is clear. For example, is the phrase "Reference transcripts with FSM-based evidence for which all associated isoforms have been removed" identical in meaning to "Reference genes with FSM-based evidence for which all reference transcript models have been removed"? If yes, the latter would avoid misinterpretations.

Thank you for pointing this out. The reviewer is right that we used "transcripts" and "isoforms" in most cases as synonyms and this might be confusing. Both concepts are indeed similar but not identical. We have gone through the text now and make "transcript models" of those cases when we stress the structure of the RNA molecule, but kept the word isoform when referring more to the alternative isoforms that a given gene can express and their biological significance.

- Lines 247-257 would benefit from repeating the total size of the FSM and ISM numbers. Otherwise, counts in this paragraph are difficult to interpret.

This section is now lines 133-144 in the revised manuscript. We provide both total number and percentages of FSM and ISM with 5' end support to make the text more clear.

- "The detection of a polyA motif was found to be significantly associated with 3' end validation by Quant-seq (p -value = 2.2×10^{-16} , Fisher's Exact Test, see Methods and Supplementary Fig.A2a)." The figure shows three distinct variables, but the phrase concentrates on two. The authors should explore different depictions focusing on the two variables mentioned in the text.

Thank you for pointing out the inconsistency between the text and the supplementary figure. We have updated the text to better reflect the information in the upset plot in Fig.A2a (now Supplementary Figure 3a) "When examining the relationship among these metrics, found most TTS (156,612) agree in all three metrics, followed by those where a polyA motif was detected

in close proximity to an annotated polyA site (25,696), and only in 1% (2,269) of the cases the Quant-seq peak overlapped an unannotated polyA site (Supplementary Figure 3a)" (lines 170-174), which supports our conclusion of the Quanti-seq limitations.

- There are some spelling mistakes, including "additionaldata" (with a missing space) in line 344.

We have corrected the spelling mistake and checked the whole document for correct spelling.

- It is unclear what the "OR configuration" is in line 355

This means that several validation criteria are considered and the filter is passed if one OR the other is fulfilled, y opposite to an AND configuration that requires that two (or more) criteria are satisfied. We have added a call to the Methods section to clarify (now in Supplementary Note 4).

- The language sometimes lets readers guess what is exactly meant. For example, lines 205 and 446 use the term "automatic recues", but in between the term is not used. In line 446, the reader can struggle. Using "FSM-based rescue" or similar would enhance readability.

We are sorry for the confusing language. The notation *automatic rescue* is used by SQANTI3 to indicate rescue of discarded FSM transcript models and as such referred in the tool's documentation. We prefer to maintain this notation. In other to improve clarity we keep the italic notation for automatic rescue throughout the text, to highlight the specific meaning it has within the SQANTI3 context.

- In the rescue section, I first interpreted the phrase "10,389 isoforms and 2,884 genes were recovered from the reference transcriptome, while 36,811 transcripts and 2,638 known genes were included on average when the rescue was applied to the rules-filtered datasets" as that all isoforms came from the rescued genes. However, this would be a disturbingly high ratio in the latter case (36,811 transcripts for 2,638 known genes). I now think that isoforms were rescued from all genes. Either way, the language should make it very clear.

The rescue section has been re-written to simplify the manuscript and reduce extension and these numbers are no long explicitly provided. In the new version of this section we instead highlight the fact that many of the artifact transcripts and genes are represented again in the transcriptome thanks to the rescue process. "After successive steps to validate the candidates and reduce this multiplicity (see Methods), 11,599 artifacts (89%) were re-incorporated into the transcriptome, 94.1% of which were assigned to a reference transcript and 5.9% were assigned to a long read-defined transcript model, with a total of 2,884 new genes being added. Notably, only 11% of artifacts remained unassigned to a suitable replacement transcript after the rescue process".

- The paragraph in lines 479-491 is difficult to read because of the very high abundance of abbreviations.

The paragraph has been rewritten and the number of abbreviations considerably reduced (lines 264-286).

- In the direct RNA section, the authors mention "51,843 known genes". Given ~20k known protein-coding genes, this phrase begs for some explanations about lncRNAs their expression levels, etc (Most likely this set is dominated by lncRNAs, which in most cases have rather few reads).

TALON pipeline for transcriptome annotation using long-reads data used the GENCODE reference annotation that includes coding and non-coding genes. We did not distinguish between them, as we are not analyzing in this section the coding potential of the transcripts, just evaluating the TALON tool. Moreover, in this section we aimed to show how SQANTI3 applies to data other than PacBio-IsoSeq3. The usage of a different tool (TALON) and data type (dRNA ONT) leads to the indicated number of detected transcripts. The value of SQANTI3 is precisely help in assessing the quality of the dataset. Out of the 51,843 genes reported by TALON in this data set, 15,229 (29,3%) are annotated in the reference as lncRNA genes, meanwhile 20,500 (39%) are protein coding genes. This enforces the bias that some tools have for the previously annotated information and how SQANTI3 can help to identify (and, if desired, correct) this situations. In fact, after SQANTI3 QC, 231,716 LRTM were cataloged as coding while 104,730 as non-coding.

Reviewer #3:

Remarks to the Author:

Pardo-Palacios and Arzalluz-Luque present SQUANTI3 software for curating of RNA isoforms using long read data. They show performance and benchmarking analyses using long read direct RNA and cDNA datasets.

This software aims to fill the gap of well-developed tools for isoform curation using long read DRS and cDNA datasets. I like that the authors incorporated orthogonal markers in this process vs. focusing on just assembling/curating sequences together.

I have a few comments:

1. I appreciate the included details of the workflow. I would recommend also including a simpler description in addition for a non-RNA-Seq experts.

We have re-drafted the paper to make it simpler. We have also included a high-level description of the tool at the end of the introductory paragraph and refer to the first panel of Figure 1. Hopefully, this will make it easier for non-RNASeq expert to understand what the tool does. This is the explanatory section:

“SQANTI3 is a tool for the quality control and annotation of long-read transcript models (LRTM). Basically, SQANTI3 compares splice-junctions (SJ), transcription start (TSS), and termination sites (TTS) of LRTM to reference transcripts to classify them as known or different types of novel transcripts. SQANTI3 also processes complementary data such as Illumina short-reads, CAGE and Quant-seq peaks to reveal orthogonal support of LRTM, and extracts over 60 features as quality indicators. Additionally, SQANTI3 supports dataset curation through flexible transcript filtering strategies that leverage the descriptive information collected by the tool. Finally, SQANTI3 also provides full functional annotation of LRTM to enable functional iso-transcriptomics analysis from these data.”

2. For the TSS/TTS variability from IsoSeq3 vs SQANTI3, did the authors validate any of these sites experimentally or computationally? That could add compelling evidence.

Thank you for your question. The experimental validation actually comes from the CAGE and Quant-seq data, which is the closest we can get to validate transcript ends. Through SQANTI3 filtering we substantially improve the confirmation of TSS and TTS by these other data types, as they are used as quality criteria.

3. How does the SQANTI3 account for the 5'-terminal missing bases in nanopore RNA sequencing?

This is an interesting question. In principle, SQANTI3 is agnostic to the sequencing platform and evaluates transcript models created by an algorithm that infers them. The algorithm should deal with this bias of nanopore and SQANTI will evaluate. However, if these are less than 50 nts, then SQANTI3 will classify this as a reference match (RM), as long as all splice-sites of the transcript are preserved.

4. Similar to 3, is there a risk in curating a transcript where the 5'-end is not unambiguously determined. I.e., what is the measure of a true 5'-cap? Is it only the orthogonal markers?

SQANTI3 uses two metrics to evaluate 5' ends, the support by orthogonal CAGE data and the TSS ratio, which we have seen is a good surrogate of CAGE support. In case the 5' end is not unambiguously determined and multiple transcript models for the same transcript are present with slightly different 5' ends, these would be validated either by multiple or broader CAGE peaks, or by a match to the reference annotation with 50 nts wiggle to account for small differences in the exact 5' end position.

5. How does SQANTI3 handle retained-introns in transcripts?

Within the transcript classification scheme, intron retentions are novel-in-catalogue transcripts and SQANTI, if the transcript has no other differences in splice-site content. In case the transcript also misses splice junctions at 3' or 5' ends, it is labelled as ISM-intron retention, or mon-exon by intron retention. We have included a sentence in the manuscript to indicate that intron retention are labelled by SQANTI (line 97).

6. The figures could be simplified, especially they feel dense and hard to read.

This remark aligns with the editor request for a simpler manuscript. We have redrafted to include only 2 figures with easy-to-interpret panels and a shorter version of the text. Most removed figures and text are included as supplementary information..

7. It would be useful to include the tutorial in GitHub, along with a couple of Jupyter notebooks.

SQANTI3 provides extensive documentation in GitHub, as celebrated by reviewer 2. However, we have created a Markdown in our GitHub Wiki including instructions to run the three SQANTI3 modules on an example dataset. You can find this tutorial at <https://github.com/ConesaLab/SQANTI3/wiki/Tutorial:-running-SQANTI3-on-an-example-dataset>

Decision Letter, first revision:

Our ref: NMETH-BC52759A

4th Dec 2023

Dear Dr. Conesa,

Thank you for submitting your revised manuscript "SQANTI3: curation of long-read transcriptomes for accurate identification of known and novel isoforms" (NMETH-BC52759A). It has now been seen by the original referees and their comments are below. The reviewers find that the paper has improved in revision, and therefore we'll be happy in principle to publish it in Nature Methods, pending minor revisions to satisfy the referees' final requests and to comply with our editorial and formatting guidelines.

TRANSPARENT PEER REVIEW

ORCID

Sincerely,
Lei

Lei Tang, Ph.D.
Senior Editor
Nature Methods

Reviewer #1 (Remarks to the Author):

The revised manuscript has been substantially simplified and streamlined, which improves the readability. In reading the response to the other reviewers' comments it is clear that SQANTI3 is primarily designed to be a data exploration and quality control tool, with the end user responsible for assessing the results. I appreciate this point, as I share the concern about users uncritically using the results leading to problems later on (e.g. artifacts making their way into annotation databases). Perhaps this should be emphasized in the abstract and introduction (for example by saying that the tool enables users to explore and make informed decisions about their data) and the 'curation' aspects should be deemphasized (the tool does not perform the curation itself, but rather /enables/ users to do their own curation, which is an important distinction in my view). The closing paragraph (lines 262-273) is a good description of limitations and intended use cases.

There is one passage in the revised manuscript that is unclear to me:

"When examining the relationship among these metrics, most TTS (156,612) agree in all three metrics, followed by those where a polyA motif was detected in close proximity to an annotated polyA site (25,696), and only in 1% (2,269) of the cases the Quant-seq peak overlapped an unannotated polyA site (Supplementary Figure 3a)."

In this context is a TTS a predicted termination site from the de novo transcriptome reconstruction, or a known TTS from the reference annotation? What does it mean that they agree on all three metrics? Supplementary Figure 3a doesn't include the "distance to the nearest same-gene annotated transcription termination site" metric that is mentioned in the previous sentence (line 168), which I find confusing.

Reviewer #2 (Remarks to the Author):

The authors have addressed my comments, as such it is almost ready to go. SQANTI3 will likely have loads of users.

One last comment, the reference to the method of the year should include references to the descriptions of the RNA portion (which is the center of this manuscript as well): Foord et al (PMID: 36635536); Lucas & Novoa (PMID: 36635536)

Reviewer #3 (Remarks to the Author):

Thank you for addressing all reviewer comments. I do not have anything further.

Author Rebuttal, first revision:

Please, find below our responses to the last comments by reviewers.

Reviewer #1:

Remarks to the Author:

The revised manuscript has been substantially simplified and streamlined, which improves the readability. In reading the response to the other reviewers' comments it is clear that SQANTI3 is primarily designed to be a data exploration and quality control tool, with the end user responsible for assessing the results. I appreciate this point, as I share the concern about users uncritically using the results leading to problems later on (e.g. artifacts making their way into annotation databases). Perhaps this should be emphasized in the abstract and introduction (for example by saying that the tool enables users to explore and make informed decisions about their data) and the 'curation' aspects should be deemphasized (the tool does not perform the curation itself, but rather /enables/ users to do their own curation, which is an important distinction in my view). The closing paragraph (lines 262-273) is a good description of limitations and intended use cases.

SQANTI3 has two main functions, quality control, which is achieved through the sqanti3.py tool, and curation, which is applied using the filtering and rescue modules. Following the reviewer's suggestion we have added the following sentences to the end of the introduction section:

The tool globally enhances user control over long-read transcriptomes, offering a comprehensive platform for systematic data exploration and curation. This empowers informed decision-making aligned with research goals and data availability

There is one passage in the revised manuscript that is unclear to me:

“When examining the relationship among these metrics, most TTS (156,612) agree in all three metrics, followed by those where a polyA motif was detected in close proximity to an annotated polyA site (25,696), and only in 1% (2,269) of the cases the Quant-seq peak overlapped an unannotated polyA site (Supplementary Figure 3a).”

In this context is a TTS a predicted termination site from the de novo transcriptome reconstruction, or a known TTS from the reference annotation? What does it mean that they agree on all three metrics? Supplementary Figure 3a doesn't include the “distance to the nearest same-gene annotated transcription termination site” metric that is mentioned in the previous sentence (line 168), which I find confusing.

We appreciate the thorough review of our manuscript and the identification of an error in the indicated paragraph, due to misspelling of the paper. In the final revised version, we have addressed this misunderstanding and improved the wording to ensure the required accuracy and clarity. Below, I present the revised version for your consideration:

Regarding 3' end diversity, SQANTI3 QC was utilized to determine the support for detected Transcript Termination Sites (TTS) through three distinct types of evidence: Quant-seq data, the polyASite database annotation \cite{herrmann_polyasite_2020}, and the presence of a polyadenylation (polyA) motif within the final 50bp of the transcript sequence. When examining the correlation among them, the majority of transcripts exhibited a TTS (165,612) supported by all three sources of evidence. This was followed by instances where a polyA motif was identified in close proximity to an annotated polyA site (25,696). In only 1% of cases (2,269), the Quant-seq peak overlapped with an unannotated polyA site (refer to Supplementary Figure 3a).

Reviewer #2:

Remarks to the Author:

The authors have addressed my comments, as such it is almost ready to go. SQANTI3 will likely have loads of users.

One last comment, the reference to the method of the year should include references to the descriptions of the RNA portion (which is the center of this manuscript as well): Foord et al (PMID: 36635536); Lucas & Novoa (PMID: 36635536)

Thank you for pointing out these references. We have added them to the manuscript.

Reviewer #3:

Remarks to the Author:

Thank you for addressing all reviewer comments. I do not have anything further.

Thank you for your useful previous comments!

Final Decision Letter:

1st Mar 2024

Dear Dr Conesa,

I am pleased to inform you that your Brief Communication, "SQANTI3: curation of long-read transcriptomes for accurate identification of known and novel isoforms", has now been accepted for publication in *Nature Methods*. The received and accepted dates will be 1st Jun 2023 and 1st Mar 2024. This note is intended to let you know what to expect from us over the next month or so, and to let you know where to address any further questions.

Over the next few weeks, your paper will be copyedited to ensure that it conforms to *Nature Methods* style. Once your paper is typeset, you will receive an email with a link to choose the appropriate publishing options for your paper and our Author Services team will be in touch regarding any additional information that may be required.

Once proofs are generated, they will be sent to you electronically and you will be asked to send a corrected version within 48 hours. It is extremely important that you let us know now whether you will be difficult to contact over the next month. If this is the case, we ask that you send us the contact information (email, phone and fax) of someone who will be able to check the proofs and deal with any last-minute problems.

If, when you receive your proof, you cannot meet the deadline, please inform us at rjsproduction@springernature.com immediately.

Please note that *Nature Methods* is a Transformative Journal (TJ). Authors may publish their research with us through the traditional subscription access route or make their paper immediately open access

through payment of an article-processing charge (APC). Authors will not be required to make a final decision about access to their article until it has been accepted. Find out more about Transformative Journals

If you are active on Twitter/X, please e-mail me your and your coauthors' handles so that we may tag you when the paper is published.

To assist our authors in disseminating their research to the broader community, our SharedIt initiative provides you with a unique shareable link that will allow anyone (with or without a subscription) to read the published article. Recipients of the link with a subscription will also be able to download and print the PDF. As soon as your article is published, you will receive an automated email with your shareable link.

Please note that you and your coauthors may order reprints and single copies of the issue containing your article through Springer Nature Limited's reprint website, which is located at <http://www.nature.com/reprints/author-reprints.html>. If there are any questions about reprints please send an email to author-reprints@nature.com and someone will assist you.

Best regards,
Lei

Lei Tang, Ph.D.
Senior Editor
Nature Methods